# Discrimination-free Pricing with Privatized Sensitive Attributes

## Abstract

Fairness has emerged as a critical consideration in the landscape of machine learning algorithms, particularly as AI continues to transform decision-making across societal domains. To ensure that these algorithms are free from bias and do not discriminate against individuals based on sensitive attributes such as gender and race, the field of algorithmic bias has introduced various fairness concepts, including demographic parity and equalized odds, along with methodologies to achieve these notions in different contexts. Despite the rapid advancement in this field, not all sectors have embraced these fairness principles to the same extent. One specific sector that merits attention in this regard is insurance. Within the realm of insurance pricing, fairness is defined through a distinct and specialized framework. Consequently, achieving fairness according to established notions does not automatically ensure fair pricing. In particular, regulatory bodies are increasingly emphasizing transparency in pricing algorithms and imposing constraints on insurance companies on the collection and utilization of sensitive consumer attributes. These factors present additional challenges in the implementation of fairness in pricing algorithms. To address these complexities and comply with regulatory demands, we propose a straightforward method for constructing fair models that align with the specific fairness criteria unique to the insurance pricing domain. Notably, our approach only relies on privatized sensitive attributes and offers statistical guarantees. Further, it does not require insurers to have direct access to sensitive attributes, and it can be tailored to accommodate varying levels of transparency as required. This methodology seeks to meet the growing demands for privacy and transparency set forth by regulators while ensuring fairness in insurance pricing practices.

## 1 Introduction

Fairness has emerged as a critical consideration in the landscape of machine learning algorithms. Various concepts of algorithmic fairness have been established in this burgeoning field including demographic parity, equalized odds, predictive parity, among others Calders et al. (2009); Dwork et al. (2011); Feldman (2015); Hardt et al. (2016); Zafar et al. (2017b); Kusner et al. (2018). Each fairness concept bears its own merits that align with specific contextual applications. For instance, equalized odds is commonly considered as a preferred fairness metric in credit lending predictions. In addition to the theoretical underpinnings of these fairness notations, the literature has also witnessed a substantial development of methodologies aimed at implementing fairness criteria Zemel et al. (2013); Zafar et al. (2017a); Calmon et al. (2017); Dwork et al. (2018); Williamson & Menon (2019).

In contrast to algorithmic fairness, the insurance industry employs a unique and specialized framework, known as actuarial fairness. This well-established concept serves as a fundamental principle in pricing insurance contracts Frees & Huang (2023). The premium is considered actuarially fair if it accurately reflects the expected cost of the coverage provided to the policyholder. Given the stringent regulatory environment, insurers are mandated to demonstrate actuarial fairness in their premiums. As machine learning algorithms become more prevalent in insurance company operations, regulatory bodies in recent years have begun to reassess the concept of fairness, in particular, questioning whether an actuarial fair premium should discriminate against policyholders based on sensitive attributes, such as gender and ethnicity. For instance, Directive 2004/113/EC ("Gender Directive") issued by the Council of the European Union (the legislature) prohibits insurance companies in the UE from using gender as a risk-rating factor for pricing insurance products Xin & Huang (2023). Due to such regulatory constraints, insurance companies are either prohibited from directly accessing sensitive

attributes or are limited to accessing only a noised version of the sensitive attributes. Under this backdrop, our research aims to develop a method enabling insurers to integrate machine learning algorithms in the context of insurance pricing while adhering to the regulatory mandates regarding fairness, transparency, and privacy. As underscored by Lindholm et al. (2022b) the actuarial fairness and algorithm fairness may not coexist simultaneously under certain conditions. Consequently, our focus is on the discrimination-free premium, a conceptual frame of actuarially fair premium recently introduced in the actuarial science literature. This discrimination-free premium, aligned with the notion of fairness from a causal inference perspective, is free from both direct and indirect discrimination linked to sensitive attributes Lindholm et al. (2020).

We consider a multi-party training framework, where the insurer possesses direct access to non-sensitive attributes of policyholders but lacks access to the true sensitive attributes. Instead, a noised or privatized version of sensitive attributes is securely stored with a trusted third party. The central premise of our method is that the insurer forwards transformed non-sensitive attributes and the response variable to the trusted third party. Subsequently, the third party combines the privatized sensitive attributes and information provided by the insurer to train a machine learning model. The resulting actuarial fair premium is then transmitted back to the insurer. In our method, the noise in sensitive attributes can arise in various scenarios including but not limited to: 1) Data collection mechanisms: In the data collection, whether conducted by the insurer or a third party, privacy mechanisms are employed as filters to encourage consumers to provide relevant information. These mechanisms introduce a degree of distortion to protect individual privacy. 2) Measurement errors: Sensitive attributes contain errors stemming from inaccuracies in the information provided by policyholders. This includes instances where policyholders furnish inaccurate information in sensitive attributes, intentionally or unintentionally. 3) Privatization for data transmission security: Sensitive attributes undergo privatization to ensure data transmission security. This may happen during transmissions from third parties to insurers or vice versa. The privatization process adds a layer of security but introduces noise in the sensitive attributes. It is crucial to emphasize that the multi-party training framework we consider is general and includes two scenarios as trivial cases. First, the insurer is able to obtain the privatized sensitive attributes from a third party and apply the proposed algorithm directly. Second, the insurer collects information on both non-sensitive and sensitive attributes and sends this information to a third-party vendor to execute the pricing algorithm. Furthermore, the proposed algorithm is readily applicable when non-sensitive attributes originate from third-party sources. However, its practical value becomes less evident, as it is common practice for insurers to acquire additional policyholder information through third parties such as credit reports. In our study, we consider two practical scenarios:

1. Known noise rate: the trusted third party has full information regarding the privatized sensitive attributes, including both the privacy mechanism and the noise rate.
2. Unknown noise rate: the trusted third party has access to the privatized sensitive attributes, with knowledge limited to the privacy mechanism and no information about the noise rate.

The proposed method enjoys several advantages: 1) The insurer need not have direct access to sensitive attributes for implementation of the method; 2) The method solely relies on the privatized sensitive attributes, irrespective of the entity responsible for gathering such information; 3) The method is straightforward to implement and provides desired statistical assurance. In the pursuit of the actuarial fairness proposed by Lindholm et al. (2020), our contributions are threefold:

1. We introduce a straightforward method for training actuarially fair models that are transparency-adaptive. Notably, the method only requires access to privatized sensitive attributes as outlined in Lemma 4.2.
2. We provide statistical assurances in scenarios where the noise rate for the privacy mechanism is known (Theorem 4.3), and extend our guarantees to situations where the noise rate is unknown (Theorem 4.5).
3. We demonstrate the empirical effectiveness of our method in practical scenarios, encompassing both cases mentioned above.

## 2 BACKGROUND AND RELATED WORK

### 2.1 FAIRNESS IN MACHINE LEARNING

In algorithmic fairness literature, researchers are mainly concerned about two kinds of fairness: 1) individual fairness Dwork et al. (2011); Barocas et al. (2019), 2) group fairness Kamishima et al.

(2012); Feldman (2015); Friedler et al. (2018). In individual fairness, similar individuals are expected to be treated similarly. For example, under some similarity measure, two similar loan applicants are expected to have a similar likelihood of loan approval. In group fairness, two loan applicants with the same profile but differ only by gender (suppose enforcing group fairness on gender) are expected to have the same likelihood of loan approval under equalized odds. Although, there are conditions under which individual fairness implies group fairness Dwork et al. (2011), two kinds of fairness are often studied separately. Our work falls into the group fairness category. Methods and algorithms to train a fair model that satisfies certain fairness notion can be divided into three categories: 1) pre-processing is one enforces fairness on the training data itself before using the data to train machine learning models Adebayo & Kagal (2016); Calmon et al. (2017); Plečko & Meinshausen (2019), 2) in-processing is to achieve a pre-specified fairness notion during training Agarwal et al. (2018; 2019); Donini et al. (2020), 3) post-processing is to enforce fairness on a trained model (possibly an unfair model) during inference Hardt et al. (2016); Woodworth et al. (2017). For post-processing method, it is essentially solving a constraint optimization problem. Although fairness is defined differently in insurance pricing, our proposed two-step procedure resembles more of a post-processing procedure but with some subtle differences due to the fact that achieving the fairness notion proposed by Lindholm et al. (2020) is not viewed as a constraint optimization problem. Therefore, one should not necessarily expect techniques in post-processing to work appropriately under insurance pricing setting. However, we still borrow some ideas from the algorithmic fairness literature: the work of Mozannar et al. (2020) inspired us the idea to recover population statistics w.r.t. true sensitive attributes using only the noisy sensitive attributes. Further, when the noise rates are not known, we modified techniques in Patrini et al. (2017) to estimate the noise rate of the privatized sensitive attributes. We would like to point out that our method is also related to learning under corrupted features. The most relevant existing works are Li et al. (2016); Al-Rubaie & Chang (2019). Al-Rubaie & Chang (2019) derived the marginalized corrupted feature (MCF) framework, but under logistic loss, one needs to minimize a surrogate loss. Li et al. (2016) proposed a regularized marginalized cross-view (RMCV) model, but restricted to the square loss. Hence, a merit of our method is that it is compatible with any valid loss function.

## 2.2 FAIRNESS IN INSURANCE PRICING

It was not until recent years that regulators started to question whether an actuarial fair premium should discriminate against policyholders based on sensitive attributes such as gender and ethnicity. This motivates the study of the reconceptualization of actuarial fairness in the actuarial science literature. Particular, on the qualitative side, Lindholm et al. (2020); Shimao & Huang (2022); Xin & Huang (2023); Frees & Huang (2023) comprehensively discussed various aspects of the concept of fairness under the context of insurance. Specifically, discrimination in insurance pricing is divided into two categories: 1) direct discrimination refers to directly using the sensitive attributes as a risk-rating factor, 2) indirect discrimination (proxy discrimination) refers to the scenario where the sensitive attributes are not used in the rating algorithm, but the price/classification is unfair due to the fact that there are proxy variables in the set of non-sensitive attributes or the sensitive attributes can be well-inferred by variables in the set of non-sensitive attributes. On the quantitative side, There are mainly three approaches to train fair pricing models: 1) counterfactual approach from causal statistics Iturria et al. (2022), 2) group fairness approach similar to that of in algorithmic fairness Grari et al. (2022), 3) probabilistic approach Lindholm et al. (2022a). We'd like to point out that all mentioned works rely on direct access to true sensitive attributes which seems to be not aligned with the increasingly stringent regulatory environment. Hence, ours is the first work that considers the real-world challenges in training discrimination-free insurance pricing models in the actuarial science literature, that is the need of direct access to true sensitive attributes is relaxed to only a noisy version of true sensitive attributes. We'd also like to point out that the work of Lindholm et al. (2022a) is similar to ours, but their framework requires the insurer to have access to the true sensitive attributes and is limited to neural networks. As mentioned in Section 1, Lindholm et al. (2022b) points out that under conditions, a discrimination-free price satisfies none of the group fairness axioms defined in algorithmic fairness (i.e. demographic parity, equalized odds, and predictive parity). Hence, this motivates studies on discrimination-free pricing algorithms that comply with regulations.

## 3 PRELIMINARIES & PROBLEM FORMULATION

Consider for now, $n$ i.i.d triplets $\{X_i, Y_i, D_i\}_{i=1}^n$ drawn from an unknown distribution $(X_i, Y_i, D_i) \sim P(X, Y, D)$, where $X_i \in \mathcal{X}$ are the input variables (non-sensitive attributes), $Y_i \in \mathcal{Y}$ are the output

(response) variable (can be continuous or discrete), and $D_i \in \mathcal{D}$ are the true sensitive attribute (discrete, e.g. gender, race, etc.). Then, according to Lindholm et al. (2020).

**Definition 3.1.** Best-estimated Price: the best-estimated price for $Y$ w.r.t. $(X, D)$ is defined as:

$$\mu(X, D) := \mathbb{E}[Y|X, D].$$

$\mu(X, D)$ is a price with direct discrimination since $D$ is explicitly used in the calculation.

**Definition 3.2.** Unawareness Price: the unawareness price for $Y$ w.r.t. $X$ is defined as:

$$\mu(X) := \mathbb{E}[Y|X].$$

Although $\mu(X)$ does not explicitly depend on $D$, but as

$$\mu(X) = \int_d \mu(X, d)d\mathbb{P}(d|X).$$

Therefore, $\mu(X)$ is a price with indirect discrimination since one can potentially infer $D$ from $X$ if they are highly correlated. $\mu(X)$ is discrimination-free w.r.t. $D$ if and only if $X \perp D$. To break the chain between $X$ and $D$ in Definition 3.2, Lindholm et al. (2020) proposed an alternative price $h^*(X)$ that is discrimination-free w.r.t. $D$.

**Definition 3.3.** Discrimination-free Price: A discrimination-free price for $Y$ w.r.t. $X$ is defined as:

$$h^*(X) := \int_d \mu(X, d)d\mathbb{P}^*(d),$$

where $\mathbb{P}^*(d)$ is defined on the same range as the marginal distribution of the sensitive attributes $D$. Then our ultimate goal is to train a model that outputs $h^*(X)$ in Definition 3.3.

## 4 DISCRIMINATION-FREE PRICING

### 4.1 DISCRIMINATION-FREE PRICING UNDER TRUE SENSITIVE ATTRIBUTION

There are two components in $h^*(X)$, namely $\mu(X, D)$ and $\mathbb{P}^*(d)$. Although a go-to choice for $\mathbb{P}^*(d)$ is the empirical marginal of $D$, it can rather be viewed as a tuning parameter to satisfy desired statistical properties (e.g. unbiasedness). Therefore, $\mu(X, D)$ is considered a much more important component in $h^*(X)$. In the following discussion, we first introduce how the multi-party training method estimates $\mu(X, D)$ and outputs $h^*(X)$ in general, and then provide a concrete example.

There are two parties participating in the process, namely the insurer and a trusted third party (TTP). $n$ i.i.d. triplets $\{X_i, Y_i, D_i\}_{i=1}^n$ are drawn from the unknown population $P(X, Y, D)$, but the insurer only observes $\{X_i, Y_i\}_{i=1}^n$, and TTP observes $\{D_i\}_{i=1}^n$. The insurer's task is to provide some transformation $T$ on $X_i$ and passes the transformed data $\{T(X_i), Y_i\}_{i=1}^n$ to TTP, this completes the first step. Now comes the second step, TTP observes $\{T(X_i), Y_i, D_i\}_{i=1}^n$, but TTP does not know the transformation $T$. TTP's task consists of two components: 1) learn $\mu(T(X), D)$, 2) output the discrimination-free price $h^*(T(X))$ using $\mu(T(X), D)$ following Definition 3.3.

Let $f_k \in \mathcal{F}, \forall k \in [|\mathcal{D}|]$, where $\mathcal{F}$ is a hypothesis class and $f_k : T(\mathcal{X}) \to \mathbb{R}_+, \forall k \in [|\mathcal{D}|]$ is a score function. Then TTP learns $\mu(T(X), D)$ by minimizing the following expected risk:

$$\mathcal{R}(f_1, \ldots, f_{|\mathcal{D}|}) = \sum_{k=1}^{|\mathcal{D}|} \mathbb{E}_{Y, T(X)|D=k}\Big[L\big(f_k(T(X)), Y\big)\Big], \tag{1}$$

then for a pre-specified $P^*(d)$, TTP computes $h^*(T(X))$ by

$$h^*(T(X)) = \sum_{k=1}^{|\mathcal{D}|} f_k(T(X)) \cdot \mathbb{P}^*(D = k). \tag{2}$$

Then, TTP sends back $\mu(T(X), D), h^*(T(X))$ and other information to the insurer upon request.

**Remarks:** This framework is quite general as the only special thing is that we learned group-specific score function $f_1, \ldots, f_k, \forall k \in [|\mathcal{D}|]$. The advantage of using group-specific score functions will be obvious when we derive the population equivalence risk using the privatized sensitive attributes (see Lemma 4.2). As one should expect, there are no restrictions on the transformation $T$ and the hypothesis class $\mathcal{F}$ in general. However, similar to the trade-off between out-of-sample performance and model complexity in general, there are intrinsic trade-offs between the model transparency and the complexity of $T$ and $\mathcal{F}$. The intuition here is that the simpler $T$ and $\mathcal{F}$, the higher model

transparency. For example, when $T$ is the identity transformation and $\mathcal{F}$ is the class of linear models, we achieve the highest model transparency as it reduces to a linear regression w.r.t. $X$ itself. Notice that compared to $T$, $\mathcal{F}$ plays a more crucial role in determining the model transparency when $\mathcal{F}$ is the class of linear models, TTP's task essentially reduces to fitting a linear regression or a logistic regression w.r.t. $T(X)$ depending on the task. Finally, notice that the group membership information $D$ is not needed in the calculation of $h^*(T(X))$. We summarize the above procedure (MPTP-D) in an algorithmic manner (see Appendix B.1).

Now, we provide a concrete example that specifies the choice of $T$ and $\mathcal{F}$.

**Example 1:** Suppose in a regression setting, let $h \in \mathcal{H}$ where $\mathcal{H}$ is a hypothesis class and $h : \mathcal{X} \to \mathbb{R}_+$ is a score function. Let $L : \mathbb{R} \times \mathbb{R} \to \mathbb{R}_+$ be a loss function, then in step one, the insurer's goal is to first minimize the following empirical risk:

$$\hat{\mathcal{R}}(h) = \sum_{i=1}^n L(h(X_i), Y_i) \tag{3}$$

using a feed-forward neural network. Suppose the neural network consists of $m$ layers, and there are $q^m$ hidden nodes in the $m^{\text{th}}$ layer. Suppose $\mathcal{X} \in \mathbb{R}^{q_0}$, let $z^{(j)} : \mathbb{R}^{q_{j-1}} \to \mathbb{R}^{q_j}, \forall j \in [m]$. Then denote the composition $z^{(m:1)} : \mathbb{R}^{q_0} \to \mathbb{R}^{q_m}$, and $T(X_i) = z^{(m:1)}(X_i)$. The insurer obtains $\{T(X_i)\}_{i=1}^n$, which is an $n \times q_m$ matrix and passes it to TTP along with $\{Y_i\}_{i=1}^n$. This completes step one.

Let $f_k \in \mathcal{F}, \forall k \in [|\mathcal{D}|]$, where $\mathcal{F}$ is the class of linear models and $f_k : T(\mathcal{X}) \to \mathbb{R}_+, \forall k \in [|\mathcal{D}|]$ is a score function. Then TTP minimizes the following empirical risk:

$$\hat{\mathcal{R}}(f_1, \ldots, f_k) = \sum_{i=1}^n \sum_{k=1}^{|\mathcal{D}|} L(f_k(T(X_i)), Y_i) \cdot \mathbf{1}\{D_i = k\}, \tag{4}$$

by fitting a linear regression w.r.t. $T(X)$. Then TTP calculates the discrimination-free price following Definition 3.3:

$$\hat{h}^*(T(X)) = \sum_{k=1}^{|\mathcal{D}|} \hat{f}_k(T(X)) \cdot \hat{\mathbb{P}}(D = k), \tag{5}$$

then return $\hat{\mu}(T(X), D), \hat{h}^*(T(X))$ to the insurer. This completes step two.

**Remark:** for binary classification task, simply modify $L : \mathbb{R} \times \{0, 1\} \to \mathbb{R}_+$, and everything else follows through. In the above example, the insurer obtains $T$ via supervised learning, but as mentioned, there is no restriction on $T$ nor on the way the insurer obtains $T$. The reason we limited $\mathcal{F}$ to linear models is to preserve some interpretability with a complex $T$. This can essentially be viewed as a trade-off between out-of-sample performance and model transparency. This is one of the reasons that GLM and GAM-based models are still the main approaches in property & casualty insurance pricing since they not only preserve transparency but also provide decent predictive power.

## 4.2 DISCRIMINATION-FREE PRICING/CLASSIFICATION UNDER NOISY SENSITIVE ATTRIBUTION WITH KNOWN NOISE PARAMETERS

We first introduce some basic knowledge about local differential privacy (LDP) and then discuss how we modify the multi-party training process to train a fair pricing model when LDP is incorporated into the data collection or data transmission referring to the interpretation of noise in Section 1. The benefit of using LDP is that the data collector does not know for certain what the true sensitive attributes are regardless the information provided is accurate or not for any observation in the data Mozannar et al. (2020). So, any model trained with this dataset is differentially private w.r.t. the sensitive attributes. The $\epsilon$-LDP mechanism $Q$ is defined as:

**Definition 4.1.**

$$\max_{s,d,d'} \frac{Q(S = d|d)}{Q(S = s|d')} \le e^\epsilon,$$

and use the randomized response mechanism in Warner (1965); Kairouz et al. (2015):

$$Q(s|d) = \begin{cases} \frac{e^\epsilon}{|\mathcal{D}| - 1 + e^\epsilon} := \pi, \text{if } s = d \\ \frac{1}{|\mathcal{D}| - 1 + e^\epsilon} := \bar{\pi}, \text{if } s \ne d, \end{cases}$$

where $|\mathcal{D}|$ denotes the cardinality of $\mathcal{D}$ and $s$ is sampled from $Q(\cdot|d)$ independently from $X$ and $Y$.

Now, we consider the first practical scenario. Similar to the setup in Section 4.1, the insurer still observes $\{X_i, Y_i\}_{i=1}^n$, provides a transformation $T$, and passes $\{T(X_i), Y_i\}_{i=1}^n$ to TTP. The difference is that now, TTP only observes the privatized sensitive attributes $S$ instead, but it knows the true conditional probabilities in the given privacy mechanism $\mathbb{P}_{s_i|d_i}$, where $S_i$ is the privatized sensitive attributes (see Definition 4.1). So the problem now becomes how TTP minimizes Eq. (1):

$$\mathcal{R}(f_1, \ldots, f_k) = \sum_{k=1}^{|\mathcal{D}|} \mathbb{E}_{Y,T(X)|D=k}\Big[L\big(f_k(T(X_i)), Y_i\big)\Big], \tag{6}$$

with access to only $S$ and then follow the same procedure in Definition 3.3 to output a fair price.

**Lemma 4.2.** *Given the privacy parameter $\epsilon$, minimizing the following risk (Risk-LDP) Eq. (7) under $\epsilon$-LDP w.r.t. privatized sensitive attributes $S$ is equivalent of minimizing Eq. (1) w.r.t. true sensitive attributes $D$ at the population level:*

$$\mathcal{R}^{LDP}(f_1, \ldots, f_k) = \sum_{k=1}^{|\mathcal{D}|} \sum_{j=1}^{|\mathcal{D}|} \mathbf{\Pi}_{kj}^{-1} \mathbb{E}_{Y,T(X)|S=j}\Big[L\big(Y, f_k(T(X))\big)\Big]. \tag{7}$$

The merit of using group-specific score functions is that with access to only $S$, a population equivalent risk Eq. (7) can be easily derived from Eq. (1). But this cannot be easily done if we replace $f_k(T(X))$ with $f(T(X), D)$. As mentioned in Section 2, existing methods that do a similar job either give a surrogate risk or are restricted to some specific loss function (see Li et al. (2016); Al-Rubaie & Chang (2019)). Empirically, TTP computes $\hat{h}^*(T(X))$ using the learned $\hat{f}_1, \ldots, \hat{f}_{|\mathcal{D}|}$ and returns $\hat{h}^*(T(X)), \hat{\mu}(T(X), D)$ and other information to the insurer upon request. We also summarize the above procedure (MPTP-S) in an algorithmic manner (see Appendix B.2). Next, we present some statistical guarantees for Risk-LDP (Eq. 7).

**Theorem 4.3.** *For any $\delta \in (0, \frac{1}{2})$, $C_1 = \frac{\pi + |\mathcal{D}| - 2}{|\mathcal{D}|\pi - 1}$, denote $VC(\mathcal{F})$ as the VC-dimension of the hypothesis class $\mathcal{F}$, and $K$ be some constant that depends on $VC(\mathcal{F})$, then under a given loss function $L : Y \times Y \rightarrow \mathbb{R}_+$, and for $f = \{f_k\}_{k=1}^{|\mathcal{D}|}$ where $f_k \in \mathcal{F}, \forall k \in [|\mathcal{D}|]$ with $f_k : T(\mathcal{X}) \rightarrow \mathbb{R}_+$ s.t. $\sup_{X \in \mathcal{X}} |f_k(T(X))| \leq M \in \mathbb{R}_+, \forall k \in [|\mathcal{D}|]$ derived from Lemma 4.2, consequently, $L(f_k(T(X), Y)) \leq \phi(M) \in \mathbb{R}_+, \forall k \in [|\mathcal{D}|], X \in \mathcal{X}, Y \in \mathcal{Y}$, where $\phi$ is some function of $M$, denote $k^* \leftarrow \arg\max_k |\hat{\mathcal{R}}^{LDP}(f_k) - \mathcal{R}^{LDP}(f_k)|$, if $n \geq \frac{8 \ln(\frac{|\mathcal{D}|}{\delta})}{\min_k \mathbb{P}(S=k)}$ then with probability $1 - 2\delta$:*

$$\hat{\mathcal{R}}^{LDP}(f) \leq \mathcal{R}(f^*) + K \sqrt{\frac{VC(\mathcal{F}) + \ln(\frac{\delta}{2})}{2n}} \frac{2C_1 \phi(M)|\mathcal{D}|}{\mathbb{P}(S=k^*)}.$$

**Remark:** In Theorem 4.3, for a fixed sample size $n$, adding more noise will result in a larger $C_1$, hence resulting in a larger generalization gap as one should expect. We introduced some boundedness on the loss, this is essentially assuming $\|T(X)\| < \infty, \forall X \in \mathcal{X}$ and the parameters of $f \in \mathcal{F}$ is also finite for any $f \in \mathcal{F}$. We'd like to argue that these are reasonable assumptions. For the boundedness of $T(X)$, this assumption has been widely made in proofs of generalization error bound (see Kakade et al. (2008)). Since the merit of our method is model interpretability, hence, assuming the finiteness of parameters gives more informative model interpretations. Notice that under $0 - 1$ loss, we do not need the above assumptions since $\phi(M) = 1, \forall M \in \bar{\mathbb{R}}$, where $\bar{\mathbb{R}}$ denotes the extended real line.

## 4.3 DISCRIMINATION-FREE PRICING/CLASSIFICATION UNDER NOISY SENSITIVE ATTRIBUTION WITH UNKNOWN NOISE PARAMETER

One might have observed that, to derive from Eq. (1) to Eq. (7) under the LDP setting, it requires knowledge about $\pi, \bar{\pi}$. However, this information might not always be accessible in practice. Therefore, we now consider the second practical scenario where the noise rates $\pi, \bar{\pi}$ are not known. The setup is the same as in Section 4.2, the only difference is that TTP does not know the true conditional probabilities $\pi, \bar{\pi}$ for the given privacy mechanism $\mathbb{P}_{s_i|d_i}$. The general idea is that TTP needs to first estimate $\pi, \bar{\pi}$ and then plug them in to compute $\hat{\mathbf{\Pi}}^{-1}$. For the remainder of this section, we introduce how TTP obtains a point estimator $\hat{\pi}$ from the data (Lemma 4.4) and state the necessary assumptions needed to establish statistical guarantees under the second practical scenario (Theorem 4.5).

**Lemma 4.4.** *Under $\epsilon$-LDP setting, with $\pi \in (\frac{1}{|\mathcal{D}|}, 1], \bar{\pi} \in [0, \frac{1}{|\mathcal{D}|})$, assuming that there exists an anchor points $\tilde{T}(X)^*$ s.t. $\mathbb{P}(D = j^*|\tilde{T}(X)^*) = 1$ for some $j^* \in [|\mathcal{D}|]$, then $\pi = \mathbb{P}(S = j^*|\tilde{T}(X)^*)$. Empirically, denote the $n$-dimension vector $\boldsymbol{\eta}_s(\tilde{T}(X)^*) = (\hat{\mathbb{P}}(S = j^*|\tilde{T}(X_1)), \ldots, \hat{\mathbb{P}}(S = j^*|\tilde{T}(X_n)))$, then $\hat{\pi} = \|\boldsymbol{\eta}_s(\tilde{T}(X)^*)\|_\infty$ and $\{\hat{P}(S = j^*|\tilde{T}(X_i))\}_{i=1}^n$ can be obtained by specifying a hypothesis class $\mathcal{G}$ and minimize $\hat{\mathcal{R}}(g) = \sum_{i=1}^n L(g(\tilde{T}(X_i)), S_i)$.*

Besides Lemma 4.4, we also need some additional assumptions to establish Theorem 4.5. Specifically, we use the following procedure to construct estimators for $C_1$ and $\pi$:

**Step 1: Grouping:** Given the observed data $\{\tilde{T}(X_i), S_i\}_{i=1}^n$, we evenly divide the data into $n_1$ groups, with $m = \frac{n}{n_1}$ samples each.

**Step 2: Estimating within groups:** for any $k \in [n_1]$, within every group $\{\tilde{T}(X_{k,j}), S_{k,j}\}_{j=1}^m$, we can derive an $m$-dimension vector $\boldsymbol{\eta}_{s,k}(\tilde{T}(X_{k,\cdot})^*) = (\hat{\mathbb{P}}_k(S = j^*|\tilde{T}(X_{k,1})), \ldots, \hat{\mathbb{P}}_k(S = j^*|\tilde{T}(X_{k,m})))$ and $\hat{\pi}_k = \|\boldsymbol{\eta}_{s,k}(\tilde{T}(X)^*)\|_\infty$, as defined in Lemma 4.4. Then, by a simple plug in $\hat{C}_{1,k} = \frac{\hat{\pi}_k + |\mathcal{D}| - 2}{|\mathcal{D}|\hat{\pi}_k - 1}$.

**Step 3: Averaging:** Finally, our estimator for $C_1$, denoted by $\hat{C}_1 = \frac{1}{n_1}\sum_{k=1}^{n_1} \hat{C}_{1,k}$, can be derived by averaging $\hat{C}_{1,k}, k \in [n_1]$.

Next, we state two assumptions that we use to derive the generalization error bound for Risk-LDP (Eq. (7)) when the noise rate is estimated from the data.

**Assumption A:** (Sub-exponentiality) For all $k \in [n_1]$, define $\hat{g}_k(\tilde{T}(X)) = \hat{\mathbb{P}}_k(S = j^*|\tilde{T}(X))$ There exists a constant $M_g > 0$, such that $\|\hat{C}_{1,k}\|_{\psi_1} = \|\min_{i\in[m]} \frac{\hat{g}_k(\tilde{T}(X_{k,i})) + |\mathcal{D}| - 2}{|\mathcal{D}|\hat{g}_k(\tilde{T}(X_{k,i})) - 1}\|_{\psi_1} \leq M_g$ for all $k \in [n_1]$, where $\|\cdot\|_{\psi_1}$ is the sub-exponential norm: $\|X\|_{\psi_1} = \inf\{t > 0|\mathbb{E}[e^{X/t}] \leq 2\}$.

**Assumption B:** (Nearly Unbiasedness) For all $k \in [n_1]$, $\hat{C}_{1,k}$ is a 'nearly' unbiased estimator of $C_1$, namely $\left|\mathbb{E}[\hat{C}_{1,k}] - C_1\right| < \theta$ for all $k \in [m]$, where $\theta > 0$.

**Remark:** According to the form of $\hat{C}_{1,k}$, the tail of this estimator is equivalent to the distribution of $\hat{\pi}_k = \max_{i\in[m]} \hat{g}_k(\tilde{T}(X_{k,i}))$ near $\frac{1}{|\mathcal{D}|}$. If $\hat{g}_k$ is a good estimator as well as $m$ is large enough, $\hat{\pi}_k$ will be concentrated near $\pi > \frac{1}{|\mathcal{D}|}$, which is guaranteed by Lemma 4.4. Especially when $\pi - \frac{1}{|\mathcal{D}|}$ is relatively large, it is reasonable to expect that $\hat{\pi}_k$ has a sparse distribution near $\frac{1}{|\mathcal{D}|}$, which implies that $\hat{C}_{1,k}$ has a sub-exponential tail (or even bounded). For Assumption B 4.3, notice that within every group $k \in [n_1]$, $\hat{\pi}_k$ are estimators for $\pi$, and thus $\hat{C}_{1,k}$ are plug-in estimator for $C_1$. Since $\hat{C}_{1,k}$ are identically and independently distributed, it is reasonable to assume $\hat{C}_{1,k}$ are 'nearly' unbiased. A more detailed discussion can be found in Appendix C. With the aforementioned assumptions, we derive the following theorem:

**Theorem 4.5.** *For any $\delta \in (0, \frac{1}{3})$, $C_1 = \frac{\pi + |\mathcal{D}| - 2}{|\mathcal{D}|\pi - 1} > 0$, $\hat{C}_1 = \frac{1}{n_1}\sum_{k=1}^{n_1} \hat{C}_{1,k}$, where $\hat{C}_{1,k}$ is defined in Lemma 4.4, denote $VC(\mathcal{F})$ as the VC-dimension of the hypothesis class $\mathcal{F}$, and $K$ be some constant that depends on $VC(\mathcal{F})$, if Assumption A (4.3),B (4.3), and Lemma 4.4 hold, given a loss function $L : Y \times Y \to \mathbb{R}_+$, $M_g + \frac{C_1 + \theta}{\ln 2} > \tilde{\epsilon} > \theta$, and for $f = \{f_k\}_{k=1}^{|\mathcal{D}|}$ where $f_k \in \mathcal{F}, \forall k \in [|\mathcal{D}|]$ with $f_k : T(\mathcal{X}) \to \mathbb{R}_+$ s.t. $\sup_{X \in \mathcal{X}} |f_k(T(X))| \leq M \in \mathbb{R}_+, \forall k \in [|\mathcal{D}|]$ derived from Lemma 4.2, consequently, $L(f_k(T(X), Y)) \leq \phi(M) \in \mathbb{R}_+, \forall k \in [|\mathcal{D}|], X \in \mathcal{X}, Y \in \mathcal{Y}$, where $\phi$ is some function of $M$, denote $k^* \leftarrow \arg\max_k |\hat{\mathcal{R}}^{LDP}(f_k) - \mathcal{R}^{LDP}(f_k)|$, if $n \geq \frac{8\ln(\frac{|\mathcal{D}|}{\delta})}{\min_k \mathbb{P}(S = k)}$, $n_1 \geq \frac{1}{c(\tilde{\epsilon} - \theta)^2}(M_g + \frac{C_1 + \theta}{\ln 2})^2 \ln(\frac{2}{\delta})$ where $c$ is an absolute constant, then with probability $1 - 3\delta$:*

$$\hat{\mathcal{R}}^{LDP}(f) \leq \mathcal{R}(f^*) + K\sqrt{\frac{VC(\mathcal{F}) + \ln(\frac{\delta}{2})}{2n}} \frac{2(C_1 + \tilde{\epsilon})\phi(M)|\mathcal{D}|}{\mathbb{P}(S = k^*)}.$$

**Remark 1:** Notices that as $n_1$ increases $\hat{C}_1$ is more accurate since it is the average of $n_1$ independent variables, hence resulting a tighter bound. However, blindly choosing a large $n_1$ is not recommended, since Assumption A 4.3 will not hold if $m = \frac{n}{n_1}$ is not large enough. Some light tuning might be helpful in the selection of $n_1$ in practice.

**Remark 2:** Generally speaking, the derived bound suffers more from the underestimation of $\pi$. Notice that the parameter that genuinely participates in the error bound is $\frac{1}{\pi-1/|\mathcal{D}|}$. Hence, when $\pi$ is close to $\frac{1}{|\mathcal{D}|}$, an underestimation of $\pi$ can much be destructive than overestimation (especially for $\hat{\pi} \leq \frac{1}{|\mathcal{D}|}$). Further, an empirical study on the effect of estimation error is given in Section 5.

## 5 EXPERIMENTS & RESULTS

We evaluate the performance of our proposed methods (MPTP-S) on two data sets and show that the experiment results are in support of our theories. We tested our method in a regression task (MSE loss) using the US Health Insurance data set. Further, we tested our method in a classification task (Cross-Entropy loss) using the Adult dataset (Becker & Kohavi (1996)), which is a commonly used data set in algorithmic fairness literature. For conciseness, we only present the results for regression task (Insurance) in this section, however, all results for classification task (Adult) can be found in Appendix F.

### 5.1 DATA & EXPERIMENTS

The US Health Insurance data set contains 1338 observations, 6 features, and 1 response. In our experiment, we choose $D = $ sex to be the sensitive attribute taking values "Male" and "Female". privatized sensitive attribute $S$ is generated under different privacy levels using a set of $\epsilon$'s by Definition 4.1. $D$ was used to set the performance benchmark and is masked under any other settings.

We conduct experiments 1) when the noise rate $\pi, \bar{\pi}$ are known (scenario 1) and 2) when the noise rates are unknown (scenario 2). For both scenarios, while we limited the hypothesis class $\mathcal{F}$ to the class of linear models, the insurer obtains two transformations $T_1, T_2$ for the main task and one transformation $\tilde{T}$ for noise rate estimation, where $T_1$ is obtained via supervised learning (same as example 4.1), $T_2, \tilde{T}$ are simply the identity. The reason that we choose such $T_1, T_2$ is to showcase the relationship between the complexity of $T$, model transparency, and performance on unseen data under the same $\mathcal{F}$. Further, under scenario 2, we set $n_1 = 1, 2, 4$ and conduct experiments for each $n_1$ respectively. With $\mathcal{F}$ being the class of linear models under both scenarios, TTP is essentially fitting a linear regression w.r.t. $T_1(X)$ and $T_2(X)$ to obtain $\mu(T_1(X), D)$, and $\mu(T_2(X), D)$ respectively. For the calculation of $h^*(T(X))$, we choose the empirical marginal of $D$ (estimated using $S$).

### 5.2 RESULTS

For each noise level, we generated $S$ using 5 different seeds, hence each figure below (Figure 1, 2, 3) shows the mean values across all 5 different seeds. For both scenarios, we run experiments over 7 different privacy levels for $\pi = (0.9, 0.8, 0.7, 0.6, 0.55, 0.525, 0.5175)$. As the focus is to estimate $\mu(X, D)$, for conciseness, plots for test loss of $h^*(X)$ are deferred to Appendix F.

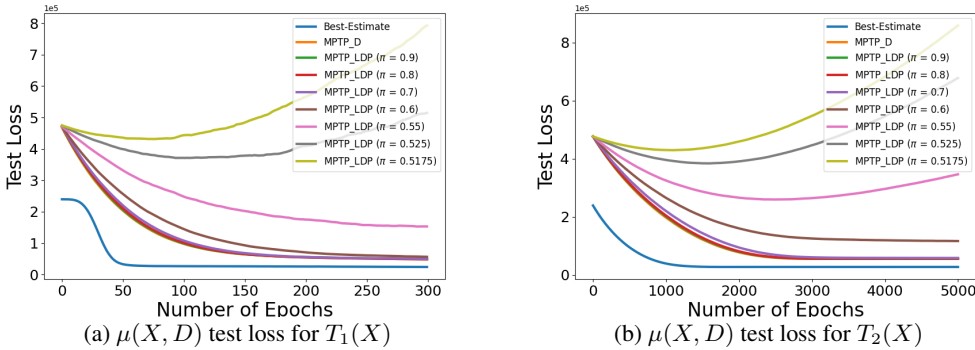

(a) $\mu(X, D)$ test loss for $T_1(X)$      (b) $\mu(X, D)$ test loss for $T_2(X)$

Figure 1: Test Loss for Scenario 1

From Figure 1, we observe that the $T_1$ is slightly more robust against noise compared to $T_2$, and $T_1$ converges much faster and has a better out-of-sample performance. Notice that as $\pi \to \frac{1}{|\mathcal{D}|}$, it

requires a larger sample size to achieve the same loss approximation. Hence, for a fixed sample size, the larger the noise, the worse Eq. (7) approximates Eq. (1) which is in support of the result we obtain from Theorem 4.3. Although in terms of both accuracy and loss, there is a gap between Eq. (1), the trade-off comes from the ease of implementation (use of group-specific models) and transparency w.r.t. $T(X)$ in that we have limited $\mathcal{F}$ to be the class of linear models. Next, we present the test loss (see Figure 2, 3) for $\mu(X, D)$ estimation using $T_1, T_2$ under scenario 2 with $n_1 = 1, 2, 4$ respectively.

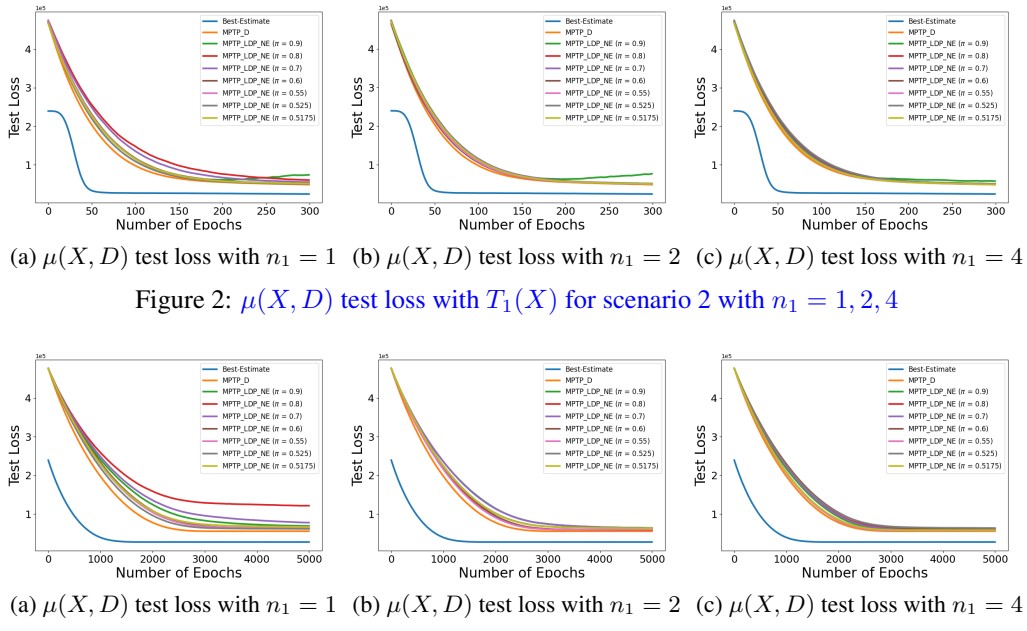

(a) $\mu(X, D)$ test loss with $n_1 = 1$    (b) $\mu(X, D)$ test loss with $n_1 = 2$    (c) $\mu(X, D)$ test loss with $n_1 = 4$

Figure 2: $\mu(X, D)$ test loss with $T_1(X)$ for scenario 2 with $n_1 = 1, 2, 4$

(a) $\mu(X, D)$ test loss with $n_1 = 1$    (b) $\mu(X, D)$ test loss with $n_1 = 2$    (c) $\mu(X, D)$ test loss with $n_1 = 4$

Figure 3: $\mu(X, D)$ test loss with $T_2(X)$ for scenario 2 with $n_1 = 1, 2, 4$

From Figure 2, 3, the loss behavior w.r.t. $T_1, T_2$ is similar to that under scenario 1 in general. However, as $n_1$ increases, we observe a better approximation of Risk-LDP (Eq. (7)) to Eq. (1) (more obvious under $T_2$). As $n_1$ increases, a smaller $\tilde{\epsilon}$ is achievable, hence resulting in a tighter bound as Theorem 4.5 suggests. Therefore, the experiment results under both scenarios align with our theoretical results.

### 5.3 Empirical Study on the Effect of Estimation Error of Noise Rate

We investigate the effect of underestimation and overestimation of $\pi$ when the distribution of sensitive attributes is imbalanced or balanced using the US Health Insurance data set by setting a group of pre-specified error of estimation ($\{\pm 0.01, \pm 0.02, \pm 0.03\}$) for each privacy level and examined the behavior of Risk-LDP (Eq. 7) for overestimation and underestimation on three manually created imbalanced subsets each with the ratio $\frac{\text{Female}}{\text{Male}} = \frac{4}{1}, \frac{2}{1}$, and $\frac{1}{1}$ respectively. Due to the page limit, we defer our investigation results in Appendix E.

## 6 Discussion & Conclusion

In this paper, we proposed a simple method to build predictors to achieve the fairness notion defined in insurance pricing that is compatible with the multi-party training framework by having a trusted third party hold information about sensitive attributes when an insurer does not have access to such information. We derived a population equivalent risk that can be optimized with access to only privatized sensitive attributes when the privatization noise rate is known and we also generalized to the setting where the noise rate is unknown. We quantified the amount of difficulty, in sample complexity that the privatization of sensitive attributes adds to the estimation of the best-estimate price. We would like to end by highlighting that future work should focus on algorithms that output discrimination-free prices w.r.t. privatized continuous variables. Further, a generalization to other kinds of privacy mechanisms is also desired.

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

# A    SUMMARY OF UPDATES

We highlighted all revisions in blue.

## A.1    UPDATE IN SECTION 1

We rearticulated some portion of the introduction section to make our work easier to follow since we believe adding more actuarial science related background to the introduction will make the audience understand better the motivation and the focus of the work. Specifically, we give definition for actuarial fair (see 1), interpretation of noise, and the generalization of the proposed method on non-sensitive attributes (see 1) (as Reviewer BixT, m8tn suggested).

## A.2    UPDATE IN SECTION 2

We further reviewed existing work on algorithms that train discrimination-free models in the actuarial science literature (see 2.2) and pointed out the value and novelty of our work (as Reviewer m8tn, BixT suggested).

## A.3    UPDATE IN SECTION 4.1

We slightly modified the example (Example 4.1) we presented to show how the choice of $T$ and $\mathcal{F}$ relates to model transparency and performance so that both regression and classification tasks are included (as Reviewer rqKi suggested).

## A.4    UPDATE IN SECTION 4.2

We relate the motivation of using LDP to the interpretation of noise to clarify that LDP is only used under two scenarios. 1) in data collection of vendors as an incentive for consumers to provide information about their sensitive attributes. 2) in data transmission for security purposes, note that 2) is not needed if the information the insurer collected is already privatized (as Reviewer m8tn suggested).

## A.5    UPDATE IN SECTION 4.3

First, we added a more technical discussion (see 4.3) on assumptions A and B (especially on the relaxation of assumption B) but deferred the presentation to Appendix C and Appendix G.4 due to the page limit (as Reviewer BixT, rqKi suggested).

Second, We added a high-level description of the impact of the estimation error of $\pi$ on the behavior of Risk-LDP (Eq. 7) (as Reviewer rqKi suggested).

## A.6    UPDATE IN SECTION 5

First, we added another empirical experiment on regression task using an insurance data set (see 5.1) to show that our method is not limited to logistic loss but is compatible with other losses as well. The results we obtained are also in support of the theoretical guarantees we derived (as Reviewer rqKi suggested).

Second, we added an empirical study on the effect of the estimation error of noise rate on both evenly and unevenly distributed scenarios. The results and observations are presented in Appendix E due to page limit (as Reviewer rqKi suggested).

# B DEFERRED ALGORITHMS

## B.1 MPTP-D

---

**Algorithm 1** Multi-party Training Process w.r.t. $D$ (MPTP-D)

---

**Insurer Input:** data: $\{X_i, Y_i\}_{i=1}^n$, hypothesis class: $\mathcal{H}$ (if obtain $T$ via supervised learning)
**Insurer Output:** $\{T(X_i)\}_{i=1}^n$
**TTP Input:** data: $\{T(X_i), Y_i, D_i\}$, hypothesis class: $\mathcal{F}$, risk function: $\mathcal{R}(f_1, \ldots, f_{|\mathcal{D}|})$ (Eq. (1))
  **repeat**
    train $f_1, \ldots, f_{|\mathcal{D}|}$ by minimizing Eq. (1)
  **until** Convergence
  compute $h^*(T(X))$ using Eq. (2)
  **return** $f_1^*, \ldots, f_{|\mathcal{D}|}^*, h^*(X)$
**TTP Output:** $f_1^*, \ldots, f_{|\mathcal{D}|}^*, h^*(T(X))$

---

## B.2 MPTP-S

---

**Algorithm 2** Multi-party Training Process w.r.t. $S$ (MPTP-S)

---

**Insurer Input:** data: $\{X_i, Y_i\}_{i=1}^n$, hypothesis class: $\mathcal{H}$ (if obtain $T$ via supervised learning), hypothesis class: $\mathcal{K}$ (if obtain $\tilde{T}$ via supervised learning)
**Insurer Output:** $\{T(X_i)\}_{i=1}^n, \{\tilde{T}(X_i)\}_{i=1}^n$
**TTP Input:** data: $\{T(X_i), Y_i, S_i\}_{i=1}^n, \{\tilde{T}(X_i), S_i\}_{i=1}^n$, hypothesis class $\mathcal{G}$, risk function: $\forall k \in [n_1], \mathcal{R}(g_k) = \sum_{j=1}^m L(g_k(\tilde{T}_{k,j}, S_{k,j}))$ (see Lemma 4.4), hypothesis class: $\mathcal{F}$, risk function: $\mathcal{R}(f_1, \ldots, f_{|\mathcal{D}|})$ (Eq. (7)),
  **if** Scenario 2 ($\pi, \bar{\pi}$ unknown) **then**
    compute $\hat{\pi}_k, \hat{\bar{\pi}}_k, k \in [n_1]$ (by applying Lemma 4.4)
    compute $\hat{C}_1$ using $\hat{\pi}_k, \hat{\bar{\pi}}_k, k \in [n_1]$ (by $C_1$ estimation procedure 4.3)
    compute $\hat{\pi}, \hat{\bar{\pi}}$ using $\hat{C}_1$
    compute $\hat{\mathbf{\Pi}}^{-1}$ using $\hat{\pi}, \hat{\bar{\pi}}$
  **else**
    compute $\hat{\mathbf{\Pi}}^{-1}$ using $\pi, \bar{\pi}$
  **end if**
  **repeat**
    train $f_1, \ldots, f_{|\mathcal{D}|}$ by minimizing Eq. (7)
  **until** convergence
  compute $h^*(T(X))$ using Eq. (2)
  **return** $f_1^*, \ldots, f_{|\mathcal{D}|}^*, h^*(X)$
**TTP Output:** $f_1^*, \ldots, f_{|\mathcal{D}|}^*, h^*(T(X))$

---

## C  DEFERRED DISCUSSION ON ASSUMPTIONS

### C.1  RESTRICTIONS ON ASSUMPTION A

The restriction of Assumption A relies on the type of generator (which will influence the tail distribution of $\hat{\pi}$) and the number of data within each group (which will influence the accuracy of $\hat{\pi}$). The condition in Assumption A is equivalent to:

$$
\mathbb{P}\left(\frac{\left(1 - \frac{1}{|\mathcal{D}|}\right)^2}{t} > \left|\hat{\pi} - \frac{1}{|\mathcal{D}|}\right|\right) \leq \exp(\frac{-t}{K}),
$$

when $K > 0$ is a constant.

Generally speaking, this assumption holds if $\hat{\pi}$ is inverse exponential distributed with a translation of $\frac{1}{|\mathcal{D}|}$, or having a lighter tail than the inverse exponential distribution that is

$$
f_{\hat{\pi}}(t) \leq \frac{1}{K(t - |\mathcal{D}|)^2}\exp(-\frac{1}{K|t - \frac{1}{\mathcal{D}}|}),
$$

when $t$ is close to $\frac{1}{|\mathcal{D}|}$, where $f_{\hat{\pi}}(t)$ is the pdf of $\hat{\pi}$. Especially, since a bounded distribution is also sub-exponential, if $|\hat{\pi} - \frac{1}{|\mathcal{D}|}| > \epsilon$, for some $\epsilon > 0$ condition is also satisfied. This will happen when the number of data within groups $(m)$ is sufficiently large and $\pi - \frac{1}{|\mathcal{D}|}$ is large enough.

### C.2  RESTRICTIONS ON ASSUMPTION B

For Assumption B, the condition is equivalent to $\mathbb{E}[\frac{1}{\hat{\pi} - 1/|\mathcal{D}|}]$. Therefore, the closer $\pi$ and $\frac{1}{|\mathcal{D}|}$ is the more accuracy of $\hat{\pi}$ is needed to suffice this assumption.

### C.3  RELAXATION OF ASSUMPTION B

We do not acquire $\hat{C}_{1,k}$ to be strictly unbiased estimators of $C_1-$ some perturbations can be allowed with some small modification to Theorem 4.5, but the general result still holds. Please see Section 4.3 and Appendix G.4 for a detailed discussion.

# D    DEFERRED EXPERIMENT RESULTS

## D.1    DATA

The Adult dataset contains $48842$ observations, $14$ features, and $1$ target variable (income). In our experiment, we delete observations with missing values which results in a subset of $45222$ observations. Further, we delete "fnlwgt", "education_num" where the former has no clear description and the latter is a duplicate of "education". We choose $D = $ sex to be our sensitive attribute which takes values of "male" and "female". The privatized sensitive attribute $S$ is generated under different privacy levels using a set of $\epsilon$'s by Definition 4.1. $D$ was only used to set the benchmark for performance and is masked under any other settings.

We conduct experiments 1) when the noise rate $\pi, \bar{\pi}$ are known (scenario 1) and 2) when the noise rates are unknown (scenario 2). For both scenarios, while we limited the hypothesis class $\mathcal{F}$ to the class of linear models, the insurer obtains two transformations $T_1, T_2$ for the main task and one transformation $\tilde{T}$ for noise rate estimation, where $T_1$ is obtained via supervised learning (same as example 4.1), $T_2, \tilde{T}$ are simply the identity. The reason that we choose such $T_1, T_2$ is to showcase the relationship between the complexity of $T$, model transparency, and performance on unseen data under the same $\mathcal{F}$. Further, under scenario 2, we set $n_1 = 1, 2, 4$ and conduct experiments for each $n_1$ respectively. With $\mathcal{F}$ being the class of linear models under both scenarios, TTP is essentially fitting a logistic regression w.r.t. $T_1(X)$ and $T_2(X)$ to obtain $\mu(T_1(X), D)$, and $\mu(T_2(X), D)$ respectively. For the calculation of $h^*(T(X))$, we choose the empirical marginal of $D$ (estimated using $S$).

## D.2    RESULTS

For each noise level, we generated $S$ using $5$ different seeds, hence each figure below (Figure 4, 5, 6) shows the mean values across all $5$ different seeds. For both scenarios, we run experiments over $7$ different privacy levels for $\pi = (0.9, 0.8, 0.7, 0.6, 0.55, 0.525, 0.5175)$. As the focus is to estimate $\mu(X, D)$, for conciseness, plots for test loss of $h^*(X)$ are deferred to Appendix F.

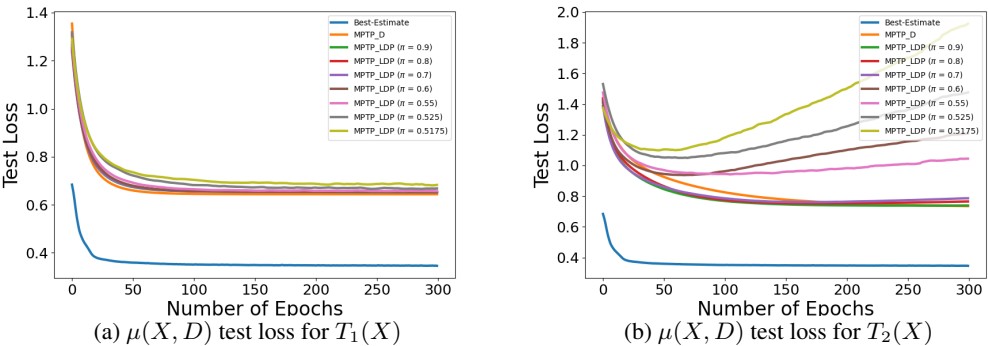

(a) $\mu(X, D)$ test loss for $T_1(X)$    (b) $\mu(X, D)$ test loss for $T_2(X)$

Figure 4: Test Loss for Scenario 1

From Figure 4, we observe that the $T_1$ is more robust against noise compared to $T_2$, and $T_1$ converges faster and has a better out-of-sample performance. Notice that as $\pi \to \frac{1}{|\mathcal{D}|}$, it requires a larger sample size to achieve the same loss approximation. Hence, for a fixed sample size, the larger the noise, the worse Eq. (7) approximates Eq. (1) which is in support of the result we obtain from Theorem 4.3. Although in terms of both accuracy and loss, there is a gap between Eq. (1), the trade-off comes from the ease of implementation (use of group-specific models) and transparency w.r.t. $T(X)$ in that we have limited $\mathcal{F}$ to be the class of linear models. Next, we present the test loss (see Figure 5, 6) for $\mu(X, D)$ estimation using $T_1, T_2$ under scenario 2 with $n_1 = 1, 2, 4$ respectively.

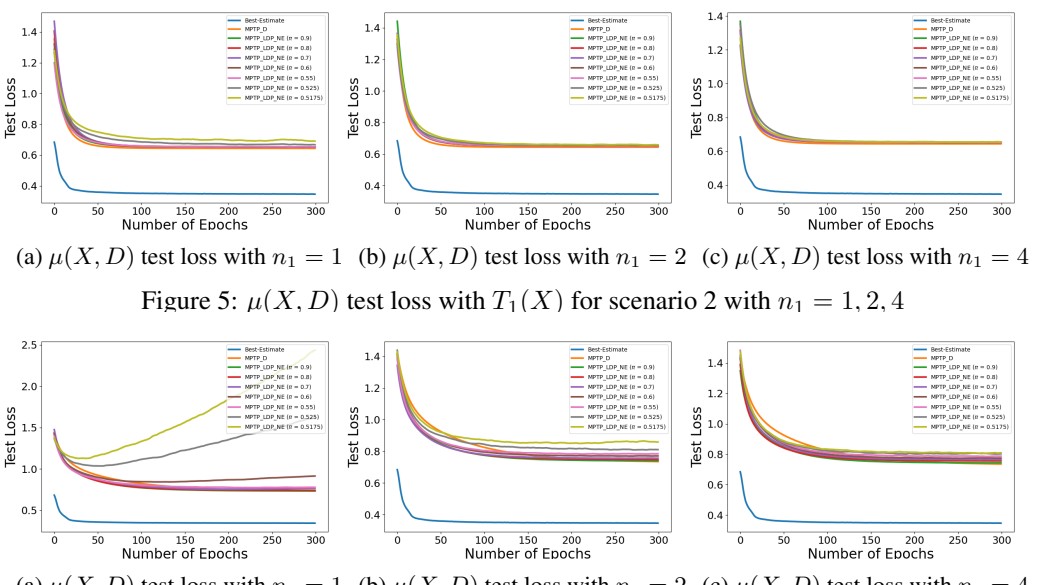

(a) $\mu(X, D)$ test loss with $n_1 = 1$  (b) $\mu(X, D)$ test loss with $n_1 = 2$  (c) $\mu(X, D)$ test loss with $n_1 = 4$

Figure 5: $\mu(X, D)$ test loss with $T_1(X)$ for scenario 2 with $n_1 = 1, 2, 4$

(a) $\mu(X, D)$ test loss with $n_1 = 1$  (b) $\mu(X, D)$ test loss with $n_1 = 2$  (c) $\mu(X, D)$ test loss with $n_1 = 4$

Figure 6: $\mu(X, D)$ test loss with $T_2(X)$ for scenario 2 with $n_1 = 1, 2, 4$

From Figure 5, 6, the loss behavior w.r.t. $T_1, T_2$ is similar to that under scenario 1 in general. However, as $n_1$ increases, we observe a better approximation of Risk-LDP (Eq. (7)) to Eq. (1) (more obvious under $T_2$). As $n_1$ increases, a smaller $\tilde{\epsilon}$ is achievable, hence resulting in a tighter bound as Theorem 4.5 suggests. Therefore, the experiment results under both scenarios align with our theoretical results.

# E    DEFERRED INVESTIGATION OF NOISE RATE ESTIMATION ERROR

We manually set the error of estimation to be $\{\pm0.01, \pm0.02, \pm0.03\}$, and our estimated noise rate is manually adjusted for each privacy level. Further, we created three subsets of the original data where the ratio between "Female" and "Male" observation is $\frac{4}{1}$, $\frac{2}{1}$ and $\frac{1}{1}$ to study the impact of the estimation error of Risk-LDP (Eq. 7) when the privatized sensitive attributes are unevenly distributed. We conduct experiments using $T_1, T_2$ with the manually adjusted erroneous noise rate on each subset respectively and compare the performance. We first present the results using $T_1$ below.

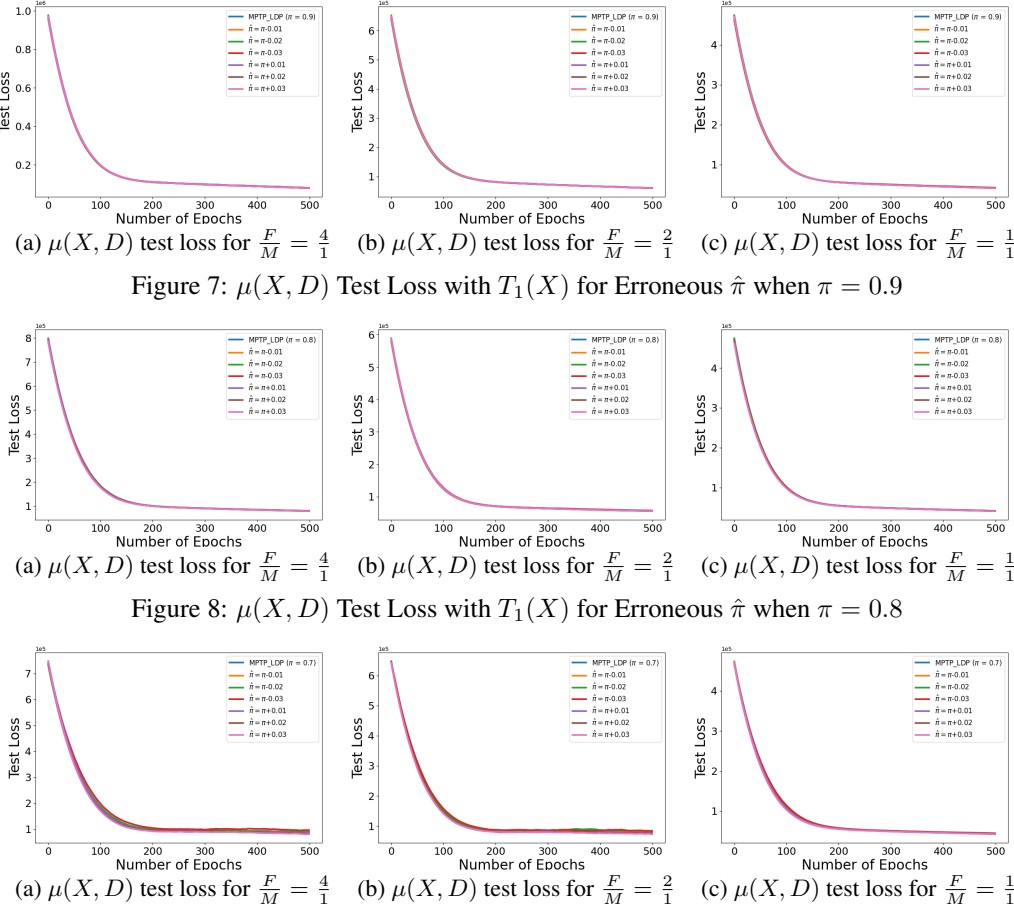

(a) $\mu(X, D)$ test loss for $\frac{F}{M} = \frac{4}{1}$    (b) $\mu(X, D)$ test loss for $\frac{F}{M} = \frac{2}{1}$    (c) $\mu(X, D)$ test loss for $\frac{F}{M} = \frac{1}{1}$

Figure 7: $\mu(X, D)$ Test Loss with $T_1(X)$ for Erroneous $\hat{\pi}$ when $\pi = 0.9$

(a) $\mu(X, D)$ test loss for $\frac{F}{M} = \frac{4}{1}$    (b) $\mu(X, D)$ test loss for $\frac{F}{M} = \frac{2}{1}$    (c) $\mu(X, D)$ test loss for $\frac{F}{M} = \frac{1}{1}$

Figure 8: $\mu(X, D)$ Test Loss with $T_1(X)$ for Erroneous $\hat{\pi}$ when $\pi = 0.8$

(a) $\mu(X, D)$ test loss for $\frac{F}{M} = \frac{4}{1}$    (b) $\mu(X, D)$ test loss for $\frac{F}{M} = \frac{2}{1}$    (c) $\mu(X, D)$ test loss for $\frac{F}{M} = \frac{1}{1}$

Figure 9: $\mu(X, D)$ Test Loss with $T_1(X)$ for Erroneous $\hat{\pi}$ when $\pi = 0.7$

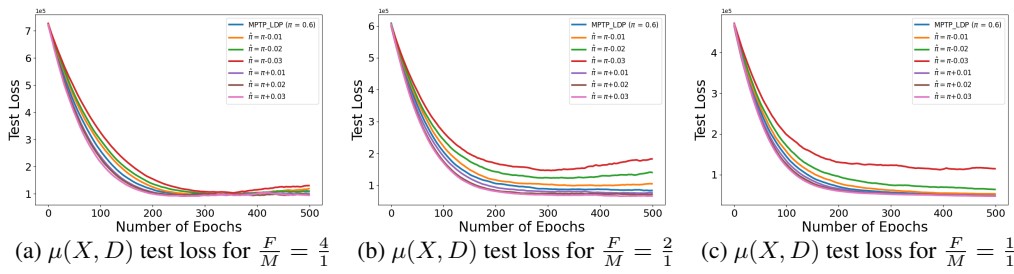

(a) $\mu(X, D)$ test loss for $\frac{F}{M} = \frac{4}{1}$    (b) $\mu(X, D)$ test loss for $\frac{F}{M} = \frac{2}{1}$    (c) $\mu(X, D)$ test loss for $\frac{F}{M} = \frac{1}{1}$

Figure 10: $\mu(X, D)$ Test Loss with $T_1(X)$ for Erroneous $\hat{\pi}$ when $\pi = 0.6$

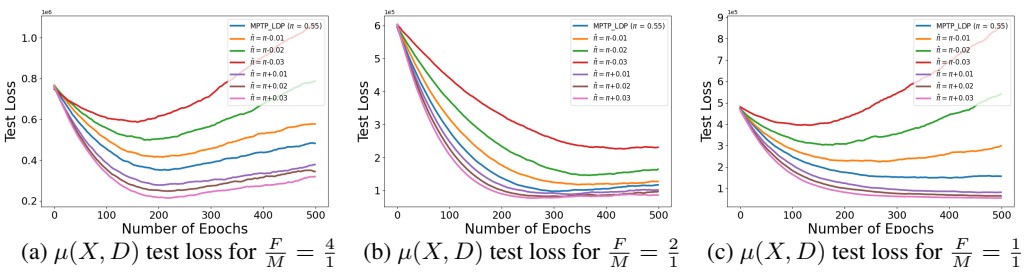

(a) $\mu(X, D)$ test loss for $\frac{F}{M} = \frac{4}{1}$    (b) $\mu(X, D)$ test loss for $\frac{F}{M} = \frac{2}{1}$    (c) $\mu(X, D)$ test loss for $\frac{F}{M} = \frac{1}{1}$

Figure 11: $\mu(X, D)$ Test Loss with $T_1(X)$ for Erroneous $\hat{\pi}$ when $\pi = 0.55$

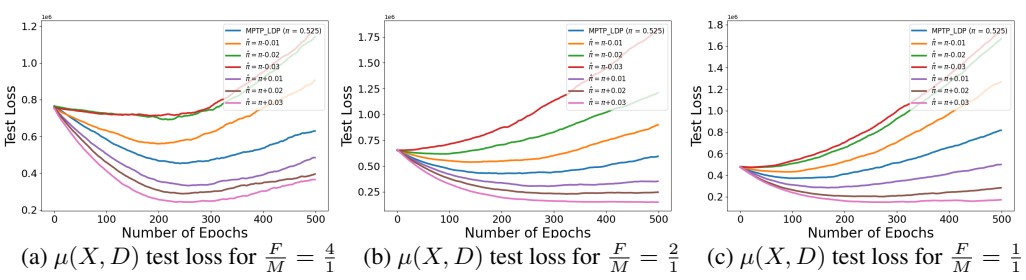

(a) $\mu(X, D)$ test loss for $\frac{F}{M} = \frac{4}{1}$    (b) $\mu(X, D)$ test loss for $\frac{F}{M} = \frac{2}{1}$    (c) $\mu(X, D)$ test loss for $\frac{F}{M} = \frac{1}{1}$

Figure 12: $\mu(X, D)$ Test Loss with $T_1(X)$ for Erroneous $\hat{\pi}$ when $\pi = 0.525$

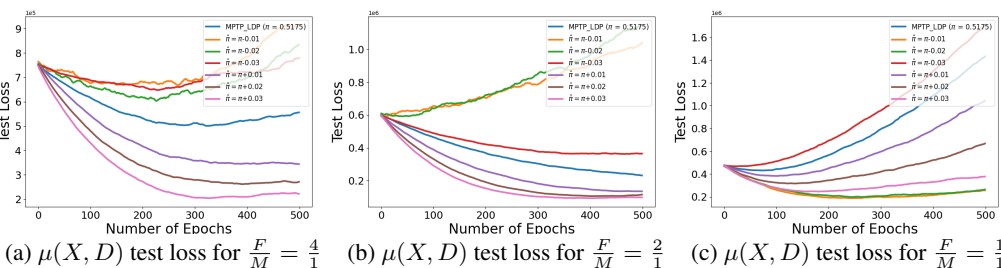

(a) $\mu(X, D)$ test loss for $\frac{F}{M} = \frac{4}{1}$    (b) $\mu(X, D)$ test loss for $\frac{F}{M} = \frac{2}{1}$    (c) $\mu(X, D)$ test loss for $\frac{F}{M} = \frac{1}{1}$

Figure 13: $\mu(X, D)$ Test Loss with $T_1(X)$ for Erroneous $\hat{\pi}$ when $\pi = 0.5175$

Next, we present the results using $T_2(X)$

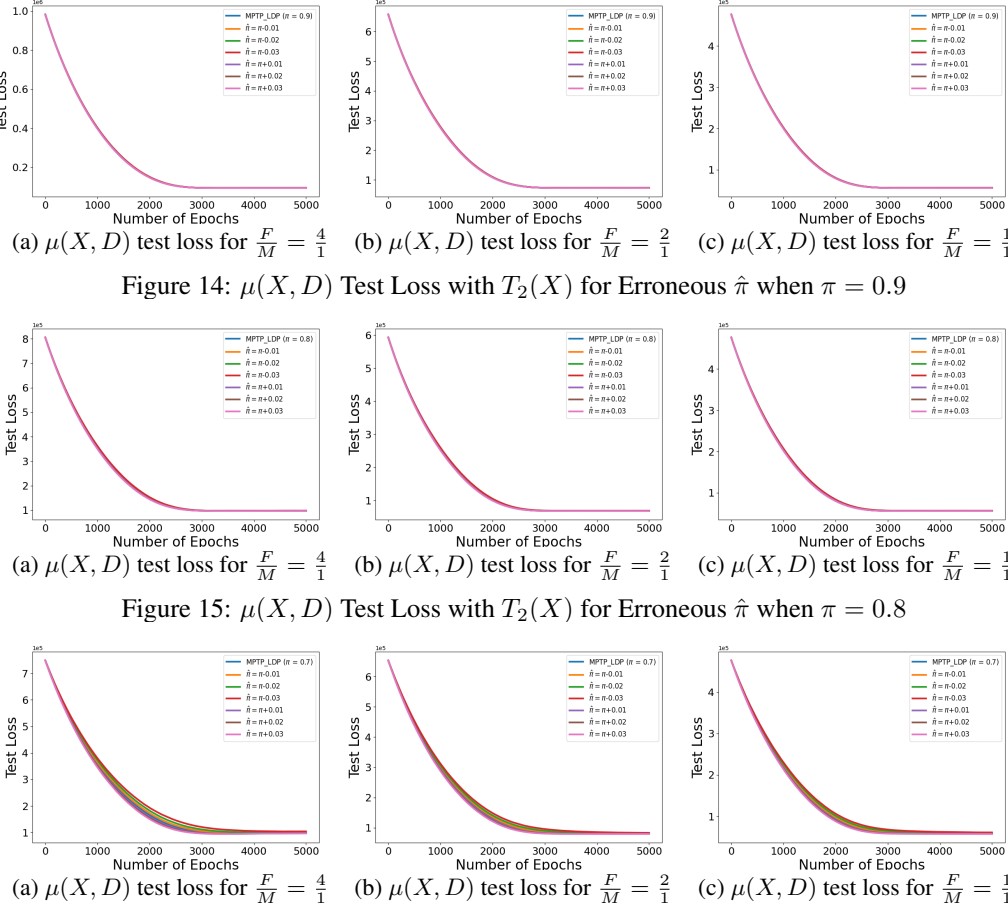

(a) $\mu(X, D)$ test loss for $\frac{F}{M} = \frac{4}{1}$ (b) $\mu(X, D)$ test loss for $\frac{F}{M} = \frac{2}{1}$ (c) $\mu(X, D)$ test loss for $\frac{F}{M} = \frac{1}{1}$

Figure 14: $\mu(X, D)$ Test Loss with $T_2(X)$ for Erroneous $\hat{\pi}$ when $\pi = 0.9$

(a) $\mu(X, D)$ test loss for $\frac{F}{M} = \frac{4}{1}$ (b) $\mu(X, D)$ test loss for $\frac{F}{M} = \frac{2}{1}$ (c) $\mu(X, D)$ test loss for $\frac{F}{M} = \frac{1}{1}$

Figure 15: $\mu(X, D)$ Test Loss with $T_2(X)$ for Erroneous $\hat{\pi}$ when $\pi = 0.8$

(a) $\mu(X, D)$ test loss for $\frac{F}{M} = \frac{4}{1}$ (b) $\mu(X, D)$ test loss for $\frac{F}{M} = \frac{2}{1}$ (c) $\mu(X, D)$ test loss for $\frac{F}{M} = \frac{1}{1}$

Figure 16: $\mu(X, D)$ Test Loss with $T_2(X)$ for Erroneous $\hat{\pi}$ when $\pi = 0.7$

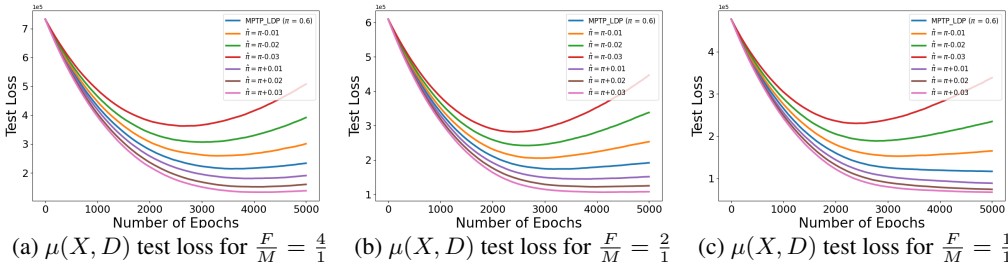

(a) $\mu(X, D)$ test loss for $\frac{F}{M} = \frac{4}{1}$    (b) $\mu(X, D)$ test loss for $\frac{F}{M} = \frac{2}{1}$    (c) $\mu(X, D)$ test loss for $\frac{F}{M} = \frac{1}{1}$

Figure 17: $\mu(X, D)$ Test Loss with $T_2(X)$ for Erroneous $\hat{\pi}$ when $\pi = 0.6$

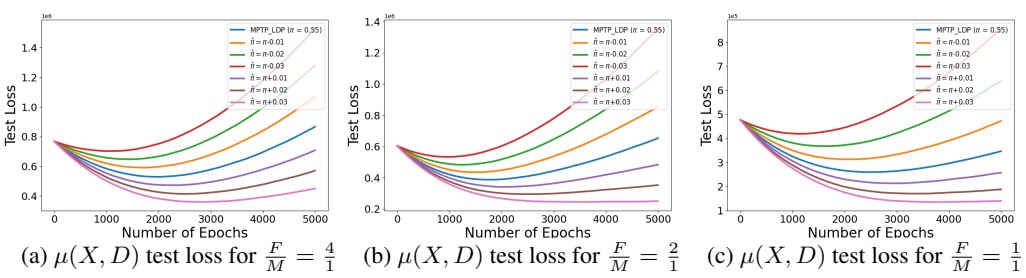

(a) $\mu(X, D)$ test loss for $\frac{F}{M} = \frac{4}{1}$    (b) $\mu(X, D)$ test loss for $\frac{F}{M} = \frac{2}{1}$    (c) $\mu(X, D)$ test loss for $\frac{F}{M} = \frac{1}{1}$

Figure 18: $\mu(X, D)$ Test Loss with $T_2(X)$ for Erroneous $\hat{\pi}$ when $\pi = 0.55$

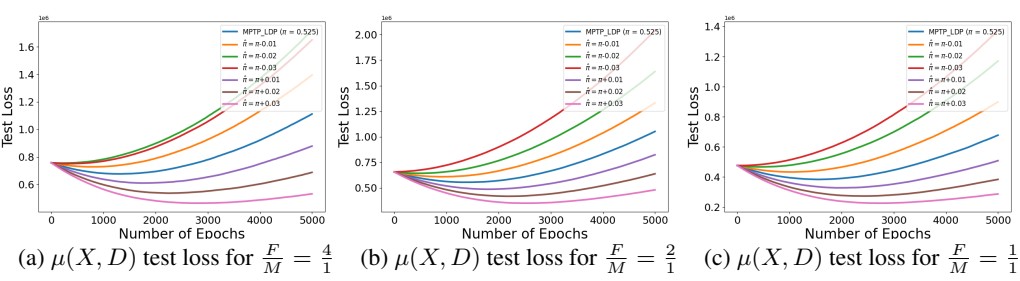

(a) $\mu(X, D)$ test loss for $\frac{F}{M} = \frac{4}{1}$    (b) $\mu(X, D)$ test loss for $\frac{F}{M} = \frac{2}{1}$    (c) $\mu(X, D)$ test loss for $\frac{F}{M} = \frac{1}{1}$

Figure 19: $\mu(X, D)$ Test Loss with $T_2(X)$ for Erroneous $\hat{\pi}$ when $\pi = 0.525$

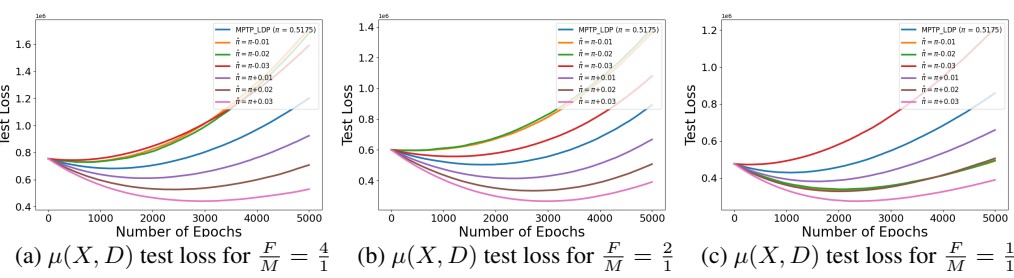

(a) $\mu(X, D)$ test loss for $\frac{F}{M} = \frac{4}{1}$    (b) $\mu(X, D)$ test loss for $\frac{F}{M} = \frac{2}{1}$    (c) $\mu(X, D)$ test loss for $\frac{F}{M} = \frac{1}{1}$

Figure 20: $\mu(X, D)$ Test Loss with $T_2(X)$ for Erroneous $\hat{\pi}$ when $\pi = 0.5175$

**Observations:** From the above figures we can see that $T_1$ converges much faster and is also more robust against estimation error on the noise rate than $T_2$ regardless of the distribution of sensitive attributes in general, combining the similar observation under scenario 1, we tend to conclude that the convergence rate regardless of the noise rate is known or unknown is considered closely related to the transformation chosen, further the robustness of Risk-LDP (Eq. 7) against noise rate estimation error is also impacted by the choice of transformation and this impact becomes more obvious as $\pi$ gets closer to $\frac{1}{|\mathcal{D}|}$.

We also observe that when $\pi$ is far away from $\frac{1}{|\mathcal{D}|}$ (in this case 0.5), regardless of the transformation chosen and the distribution of sensitive attribute, Risk-LDP (Eq. 7) is very robust against the estimation error (even when the error is large regardless overestimation or underestimation). However, as $\pi$ becomes very close to $\frac{1}{|\mathcal{D}|}$ (Figure 11, 12, 13), we can see that underestimation of $\pi$ is much more destructive than overestimation especially when the underestimation error is large.

As one should expect that different transformations should yield different behaviors of Risk-LDP. Fixing the transformation, we also noted that the distribution of the sensitive attributes does not have too much impact on the convergence behavior of Risk-LDP (Eq. 7). However, as the distribution becomes more imbalanced, Risk-LDP tends to give a higher loss than that of the less imbalanced scenario.

# F  DEFERRED FIGURES

## F.1  INSURANCE

We now present the test loss for the estimation of $h^*(X)$ using $T_1, T_2$ under scenario 1 respectively:

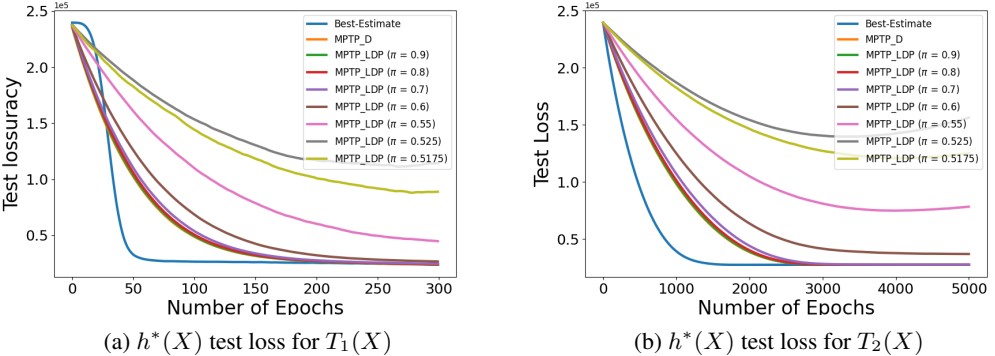

(a) $h^*(X)$ test loss for $T_1(X)$

(b) $h^*(X)$ test loss for $T_2(X)$

Figure 21: $h^*(X)$ test loss for scenario 1

Below is the test loss for $h^*(X)$ using $T_1, T_2$ with estimated $\pi$ with $n_1 = 1, 2, 4$ under scenario 2 respectively:

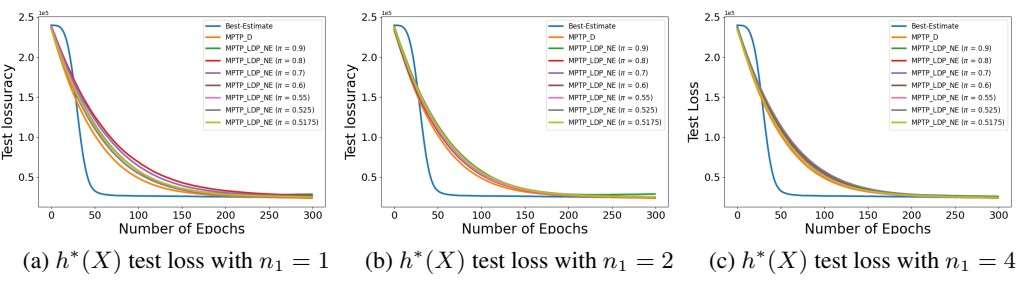

(a) $h^*(X)$ test loss with $n_1 = 1$

(b) $h^*(X)$ test loss with $n_1 = 2$

(c) $h^*(X)$ test loss with $n_1 = 4$

Figure 22: $h^*(X)$ test loss with $T_1(X)$ for scenario 2 with $n_1 = 1, 2, 4$

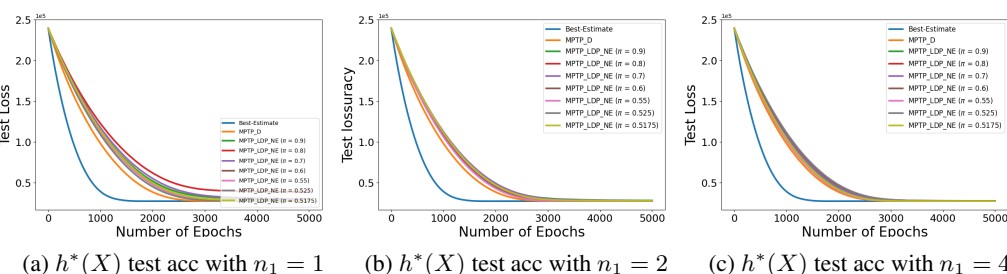

(a) $h^*(X)$ test acc with $n_1 = 1$

(b) $h^*(X)$ test acc with $n_1 = 2$

(c) $h^*(X)$ test acc with $n_1 = 4$

Figure 23: $h^*(X)$ test loss with $T_2(X)$ for scenario 2 with $n_1 = 1, 2, 4$

### F.2  ADULT

We first present test accuracy for the estimation of $\mu(X, D)$ and $h^*(X)$ using $T_1, T_2$ under scenario 1 respectively:

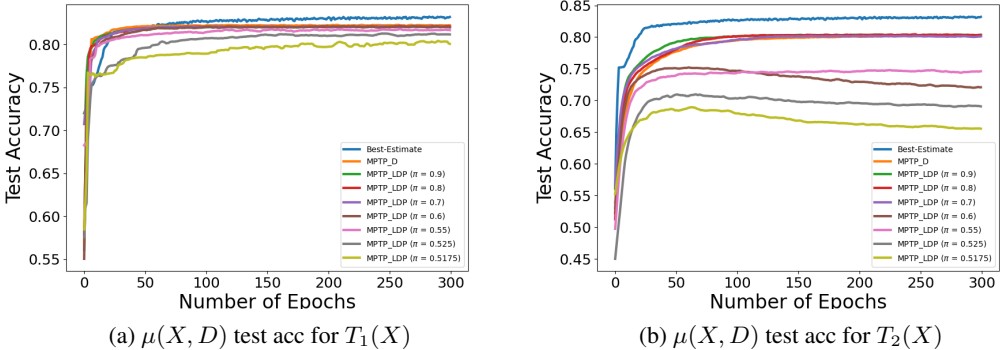

(a) $\mu(X, D)$ test acc for $T_1(X)$       (b) $\mu(X, D)$ test acc for $T_2(X)$

Figure 24: $\mu(X, D)$ test accuracy for scenario 1

Below is the test accuracy for the estimation of $\mu(X, D)$ and $h^*(X)$ using $T_1, T_2$ with $n_1 = 1, 2, 4$ under scenario 2 respectively:

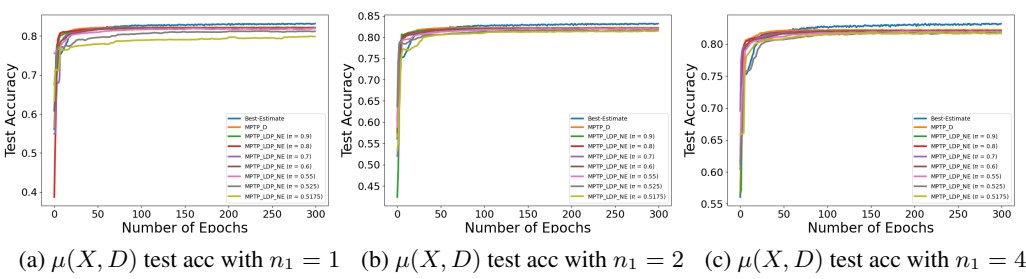

(a) $\mu(X, D)$ test acc with $n_1 = 1$   (b) $\mu(X, D)$ test acc with $n_1 = 2$   (c) $\mu(X, D)$ test acc with $n_1 = 4$

Figure 25: $\mu(X, D)$ test accuracy with $T_1(X)$ for scenario 2 with $n_1 = 1, 2, 4$

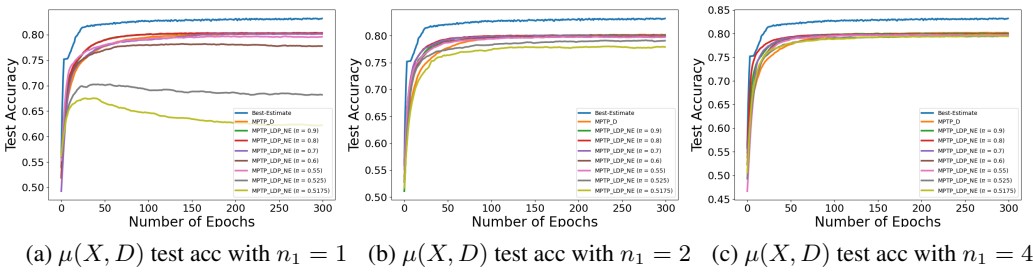

(a) $\mu(X, D)$ test acc with $n_1 = 1$   (b) $\mu(X, D)$ test acc with $n_1 = 2$   (c) $\mu(X, D)$ test acc with $n_1 = 4$

Figure 26: $\mu(X, D)$ test accuracy with $T_2(X)$ for scenario 2 with $n_1 = 1, 2, 4$

Next, we present the test accuracy for the estimation of $h^*(X)$ using $T_1, T_2$ with $n_1 = 1, 2, 4$ under scenario 2 respectively:

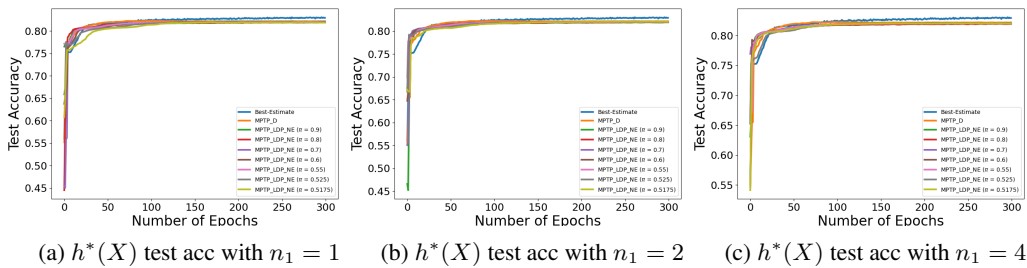

(a) $h^*(X)$ test acc with $n_1 = 1$    (b) $h^*(X)$ test acc with $n_1 = 2$    (c) $h^*(X)$ test acc with $n_1 = 4$

Figure 27: $h^*(X)$ test accuracy with $T_1(X)$ for scenario 2 with $n_1 = 1, 2, 4$

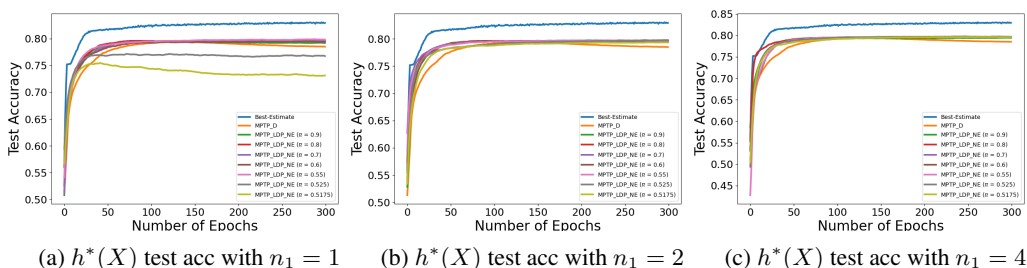

(a) $h^*(X)$ test acc with $n_1 = 1$    (b) $h^*(X)$ test acc with $n_1 = 2$    (c) $h^*(X)$ test acc with $n_1 = 4$

Figure 28: $h^*(X)$ test accuracy with $T_2(X)$ for scenario 2 with $n_1 = 1, 2, 4$

# G    DEFERRED PROOFS

## G.1    LEMMA 4.2 PROOF

**Lemma 4.2** Given the privacy parameter $\epsilon$, minimizing the following risk (Risk-LDP) Eq. (7) under $\epsilon$-LDP w.r.t. privatized sensitive attributes $S$ is equivalent of minimizing Eq. (1) w.r.t. true sensitive attributes $D$ at the population level:

$$\mathcal{R}^{LDP}(f_1,\ldots,f_k) = \sum_{k=1}^{|\mathcal{D}|}\sum_{j=1}^{|\mathcal{D}|}\mathbf{\Pi}_{kj}^{-1}\mathbb{E}_{Y,T(X)|S=j}\Big[L\big(Y, f_k(T(X)))\big)\Big], \tag{8}$$

*Proof.* **Step 1:**

Since the $\epsilon$-LDP randomization mechanism is independent of $X, Y$, therefore, the distribution of $S$ is fully characterized by the privacy parameter $\epsilon$ and the distribution of $D$. Therefore, the distribution of $S$ is deterministic once the privacy parameter $\epsilon$ and the distribution of $D$ is given.

**Step 2:** Recover distributions w.r.t. $D$

Inspired by proposition 1 in Mozannar et al. (2020). Let $\mathcal{E}_1, \mathcal{E}_2$ be two probability events defined with respect to $(T(X), Y, \hat{Y})$, then consider the following probability:

$$\begin{aligned}
&\mathbb{P}(\mathcal{E}_1, \mathcal{E}_2 \mid S = d)\\
&= \sum_{d' \in D}\mathbb{P}(\mathcal{E}_1, \mathcal{E}_2 \mid S = d, D = d')\mathbb{P}(D = d' \mid S = d)\\
&= \sum_{d' \in D}\mathbb{P}(\mathcal{E}_1, \mathcal{E}_2 \mid D = d')\mathbb{P}(D = d' \mid S = d)\\
&= \sum_{d' \in D}\mathbb{P}(\mathcal{E}_1, \mathcal{E}_2 \mid D = d')\frac{\mathbb{P}(S = d \mid D = d')\mathbb{P}(D = d')}{\sum_{d'' \in D}\mathbb{P}(S = d \mid D = d'')\mathbb{P}(D = d'')}\\
&= P(\mathcal{E}_1, \mathcal{E}_2 \mid D = d)\frac{\pi\mathbb{P}(D = d)}{\pi\mathbb{P}(D = d) + \sum_{d''\backslash d}\bar{\pi}\mathbb{P}(D = d'')} + \sum_{d'\backslash d}P(\mathcal{E}_1, \mathcal{E}_2 \mid D = d')\frac{\bar{\pi}\mathbb{P}(D = d')}{\pi\mathbb{P}(D = d) + \sum_{d''\backslash d}\bar{\pi}\mathbb{P}(D = d'')}.
\end{aligned}$$

Then, let $\mathcal{E}_1 = Y, \mathcal{E}_2 = T(X)$, we obtain the following:

$$\begin{aligned}
&\mathbb{P}(Y, T(X) \mid S = d)\\
&= \sum_{d' \in D}\mathbb{P}(Y, T(X) \mid S = d, D = d')\mathbb{P}(D = d' \mid S = d)\\
&= \sum_{d' \in D}\mathbb{P}(Y, T(X) \mid D = d')\mathbb{P}(D = d' \mid S = d)\\
&= \sum_{d' \in D}\mathbb{P}(Y, T(X) \mid D = d')\frac{\mathbb{P}(S = d \mid D = d')\mathbb{P}(D = d')}{\sum_{d'' \in D}\mathbb{P}(S = d \mid D = d'')\mathbb{P}(D = d'')}\\
&= P(Y, T(X) \mid D = d)\frac{\pi\mathbb{P}(D = d)}{\pi\mathbb{P}(D = d) + \sum_{d''\backslash d}\bar{\pi}\mathbb{P}(D = d'')} + \sum_{d'\backslash d}P(Y, T(X) \mid D = d')\frac{\bar{\pi}\mathbb{P}(D = d')}{\pi\mathbb{P}(D = d) + \sum_{d''\backslash d}\bar{\pi}\mathbb{P}(D = d'')}.
\end{aligned}$$

Denote $p_d = \mathbb{P}(D = d)$, then let $\mathbf{\Pi}$ be the following $|\mathcal{D}| \times |\mathcal{D}|$ matrix with the following entries:

$$\begin{cases}
\mathbf{\Pi}_{i,i} = \frac{\pi p_i}{\pi p_i + \sum_{d''\backslash i}\bar{\pi}p_{d''}}, \text{ for } i \in D\\
\mathbf{\Pi}_{i,j} = \frac{\bar{\pi}p_j}{\pi p_i + \sum_{d''\backslash i}\bar{\pi}p_{d''}}, \text{ for } i, j \in D \text{ s.t.}, i \neq j
\end{cases},$$

then we have the following system of linear equations:

$$
\begin{bmatrix}
\mathbb{P}(Y, T(X) \mid S = 1) \\
. \\
. \\
. \\
\mathbb{P}(Y, T(X) \mid S = |\mathcal{D}|)
\end{bmatrix}
= \boldsymbol{\Pi}
\begin{bmatrix}
\mathbb{P}(Y, T(X) \mid D = 1) \\
. \\
. \\
. \\
\mathbb{P}(Y, T(X) \mid D = |\mathcal{D}|)
\end{bmatrix},
$$

denote as $\boldsymbol{s_1} = \boldsymbol{\Pi d_1}$, where $\boldsymbol{s_1} = \mathbb{P}(Y, T(X) \mid S)$, $\boldsymbol{d_1} = \mathbb{P}(Y, T(X) \mid D)$.

Since $\boldsymbol{\Pi}$ is row-stochastic and invertible, we show that the entries of $\boldsymbol{\Pi}^{-1}$ take the following forms:

$$
\begin{cases}
\boldsymbol{\Pi}_{i,i}^{-1} = \frac{\pi + |\mathcal{D}| - 2}{|\mathcal{D}|\pi - 1} \frac{\pi p_i + \sum_{d'' \setminus i} \bar{\pi} p_{d''}}{p_i}, \text{ for } i \in D \\
\boldsymbol{\Pi}_{i,j}^{-1} = \frac{\pi - 1}{|\mathcal{D}|\pi - 1} \frac{\bar{\pi} p_i + \sum_{d'' \setminus i} \pi p_{d''}}{p_i}, \text{ for } i, j \in D \text{ s.t.,} i \neq j
\end{cases},
$$

multiplying $\boldsymbol{\Pi}^{-1}$ on both side, we recovered

$$
\mathbb{P}(Y, T(X) \mid D = k) = \sum_{j=1}^{|\mathcal{D}|} \boldsymbol{\Pi}_{kj}^{-1} \mathbb{P}(Y, T(X) \mid S = j)
$$
$$
= \boldsymbol{\Pi}_{k.}^{-1} \mathbb{P}(Y, T(X) \mid S)
$$

where $\boldsymbol{\Pi}_{k.}^{-1}$ denotes the $k^{\text{th}}$ row of $\boldsymbol{\Pi}^{-1}$.

However, there is still one component that we do need to estimate in order to recover the population distribution of $\mathbb{P}(Y, X \mid D)$. We need to further estimate $\mathbb{P}(D = d)$. Using the same technique, to estimate $\mathbb{P}(D = d)$, first write $P(S = d)$ in terms of the conditional probability of $S$ given $D$ as:

$$
\mathbb{P}(S = d) = \sum_{d' \in D} \mathbb{P}(S = d \mid D = d') \mathbb{P}(D = d')
$$
$$
= \mathbb{P}(S = d \mid D = d) \mathbb{P}(D = d) + \sum_{d' \setminus d} \mathbb{P}(S = d \mid D = d') \mathbb{P}(D = d')
$$
$$
= \pi p_d + \sum_{d' \setminus d} \bar{\pi} p_{d'}.
$$

Then we write the above expression in terms of a system of linear equations. Let $\boldsymbol{T}$ be an $|\mathcal{D}| \times |\mathcal{D}|$ matrix with the following entries:

$$
\begin{cases}
\boldsymbol{T}_{i,i} = \pi, \text{ for } i \in D \\
\boldsymbol{T}_{i,j} = \bar{\pi}, \text{ for } i, j \in D \text{ s.t.,} i \neq j
\end{cases},
$$

then we have the following system of linear equations:

$$
\begin{bmatrix}
\mathbb{P}(S = 1) \\
. \\
. \\
. \\
\mathbb{P}(S = |\mathcal{D}|)
\end{bmatrix}
= \boldsymbol{T}
\begin{bmatrix}
\mathbb{P}(D = 1) \\
. \\
. \\
. \\
\mathbb{P}(D = |\mathcal{D}|)
\end{bmatrix},
$$

denote as $\boldsymbol{s_2} = \boldsymbol{T d_2}$, where $\boldsymbol{s_2} = \mathbb{P}(S)$ and $\boldsymbol{d_2} = \mathbb{P}(D)$.

It follows the same argument that $\boldsymbol{T}$ is row-stochastic and invertible and it is easy to verify that $\boldsymbol{T}^{-1}$ takes the following form:

$$
\begin{cases}
\boldsymbol{T}_{i,i}^{-1} = \frac{\pi + |\mathcal{D}| - 2}{|\mathcal{D}|\pi - 1}, \text{ for } i \in D \\
\boldsymbol{T}_{i,j}^{-1} = \frac{\pi - 1}{|\mathcal{D}|\pi - 1}, \text{ for } i, j \in D \text{ s.t.,} i \neq j
\end{cases},
$$

by multiplying $\boldsymbol{T}^{-1}$ on both side, we obtain:

$$
\mathbb{P}(D = k) = \sum_{j=1}^{|\mathcal{D}|} \boldsymbol{T}_{kj} \mathbb{P}(S = j)
$$
$$
= \boldsymbol{T}_{k.}^{-1} \mathbb{P}(S).
$$

**Step 3:** Recover the loss w.r.t. D

At the population level, we have recovered that:

$$\mathbb{P}(Y, T(X) \mid D = k) = \boldsymbol{\Pi}_{k\cdot}^{-1} \mathbb{P}(Y, T(X) \mid S),$$

where $\mathbb{P}(D = k) = \boldsymbol{T}_{k\cdot}^{-1} \mathbb{P}(S)$ is used in calculation of $\boldsymbol{\Pi}_{k\cdot}^{-1}$.

Hence, we recover the population equivalent of Eq. (1):

$$
\begin{aligned}
\sum_{k=1}^{|\mathcal{D}|} \mathbb{E}_{Y,T(X)|D=k}\Big[L\big(Y, f_k(T(X))\big)\Big] &= \sum_{k=1}^{|\mathcal{D}|} \int_Y \int_{T(X)} \mathbb{P}(Y, T(X) \mid D = k) L\big(Y, f_k(T(X))\big) dT(X) dY \\
&= \sum_{k=1}^{|\mathcal{D}|} \int_Y \int_{T(X)} \sum_{j=1}^{|\mathcal{D}|} \boldsymbol{\Pi}_{kj}^{-1} \mathbb{P}(Y, T(X) \mid S = j) L\big(Y, f_k(T(X))\big) dT(X) dY \\
&= \sum_{k=1}^{|\mathcal{D}|} \sum_{j=1}^{|\mathcal{D}|} \int_Y \int_{T(X)} \boldsymbol{\Pi}_{kj}^{-1} \mathbb{P}(Y, T(X) \mid S = j) L\big(Y, f_k(T(X))\big) dT(X) dY \\
&= \sum_{k=1}^{|\mathcal{D}|} \sum_{j=1}^{|\mathcal{D}|} \boldsymbol{\Pi}_{kj}^{-1} \mathbb{E}_{Y,T(X)|S=j}\Big[L\big(Y, f_k(T(X))\big)\Big].
\end{aligned}
$$

Therefore, we conclude that it is equivalent to minimizing:

$$(f_{1^*}, \ldots, f_{k^*}) \leftarrow \arg\min_{f_1, \ldots, f_k} \sum_{k=1}^{|\mathcal{D}|} \sum_{j=1}^{|\mathcal{D}|} \boldsymbol{\Pi}_{kj}^{-1} \mathbb{E}_{Y,T(X)|S=j}\Big[L\big(Y, f_k(T(X))\big)\Big]$$

This completes the proof. $\qquad\square$

### G.2 LEMMA 4.4 PROOF

**Lemma 4.4** Under $\epsilon$-LDP setting, with $\pi \in (\frac{1}{|\mathcal{D}|}, 1], \bar{\pi} \in [0, \frac{1}{|\mathcal{D}|})$, assuming that there exists an anchor points $\tilde{T}(X)^*$ s.t. $\mathbb{P}(D = j^* | \tilde{T}(X)^*) = 1$ for some $j^* \in [|\mathcal{D}|]$, then $\pi = \mathbb{P}(S = j^* | \tilde{T}(X)^*)$. Empirically, denote the $n$-dimension vector $\boldsymbol{\eta}_s(\tilde{T}(X)^*) = \big(\hat{\mathbb{P}}(S = j^* | \tilde{T}(X_1)), \ldots, \hat{\mathbb{P}}(S = j^* | \tilde{T}(X_n))\big)$, then $\hat{\pi} = \|\boldsymbol{\eta}_s(\tilde{T}(X)^*)\|_\infty$ and $\{\hat{P}(S = j^* | \tilde{T}(X_i))\}_{i=1}^n$ can be obtained by specifying a hypothesis class $\mathcal{G}$ and minimize the following empirical risk:

$$\hat{\mathcal{R}}(k) = \sum_{i=1}^n L\big(k(\tilde{T}(X_i)), S_i\big).$$

*Proof.* Notice that $\pi \in (\frac{1}{|\mathcal{D}|}, 1], \bar{\pi} \in [0, \frac{1}{|\mathcal{D}|})$ and consequently we have $\pi > \bar{\pi}$. Hence, by Theorem 5 of Zhang et al. (2021), we are in a good position to apply the noise rate estimation method (Theorem 3) in Patrini et al. (2017) to estimate $\pi, \bar{\pi}$. Our $\epsilon$-LDP setting can be considered as a special case of CCN (class conditional noise) where the flip probability is the same across all groups in $\mathcal{D}$. Consider

$$
\begin{aligned}
\mathbb{P}(S = j^* | \tilde{T}(X)^*) &= \sum_{k=1}^{|\mathcal{D}|} \mathbb{P}(S = j^* | D = k) \cdot \mathbb{P}(D = k | \tilde{T}(X)^*) \\
&\overset{(a)}{=} \sum_{k=1}^{|\mathcal{D}|} \mathbb{P}(S = j^* | D = k) \cdot \mathbf{1}\{j^* = k\} \\
&= \pi,
\end{aligned}
$$

(a) is by followed by the definition of anchor point

$$\mathbb{P}(D = j^* | \tilde{T}(X)^*) = 1 \implies \mathbb{P}(D = k | \tilde{T}(X)^*) = 0, \forall k \neq j^*, k, j^* \in [|\mathcal{D}|].$$

Then one can easily see that $\mathbb{P}(S = j^*|\tilde{T}(X_i))$ attains its maximum when $\mathbb{P}(D = j^*|\tilde{T}(X_i)) = 1$, since we know

$$\begin{cases} \mathbb{P}(S = j^*|D = k) = \pi, \text{ if } j^* = k \\ \mathbb{P}(S = j^*|D = k) = \bar{\pi}, \text{ if } j^* \neq k, \end{cases}$$

hence we know $\mathbb{P}(S = j^*|\tilde{T}(X_i))$ is actually a weighted sum of $\pi$ and $\bar{\pi}$, where the weights are simply $\{\mathbb{P}(D = k|\tilde{T}(X_i))\}_{k=1}^{|\mathcal{D}|}$. But we also know that $\pi > \bar{\pi}$. Hence, for empirical estimation, denote the $n$-dimension probability vector $\boldsymbol{\eta}_s(\tilde{T}(X)^*) = \left(\hat{\mathbb{P}}(S = j^*|\tilde{T}(X_1)), \ldots, \hat{\mathbb{P}}(S = j^*|\tilde{T}(X_n))\right)$

$$\hat{\pi} = \|\boldsymbol{\eta}_s(\tilde{T}(X)^*)\|_\infty.$$

Where $\{\hat{P}(S = j^*|\tilde{T}(X_i))\}_{i=1}^n$ can be obtained by specifying a hypothesis class $\mathcal{K}$ and minimize the following empirical risk:

$$\hat{\mathcal{R}}(k) = \sum_{i=1}^n L\big(k(\tilde{T}(X_i)), S_i\big).$$

This completes the proof $\qquad\qquad\qquad\qquad\qquad\qquad\qquad\qquad\qquad\qquad\qquad\quad \square$

### G.3 THEOREM 4.3 PROOF

**Theorem 4.3** For any $\delta \in (0, \frac{1}{2})$, $C_1 = \frac{\pi + |\mathcal{D}| - 2}{|\mathcal{D}|\pi - 1}$, denote $VC(\mathcal{F})$ as the VC-dimension of the hypothesis class $\mathcal{F}$, and $K$ be some constant that depends on $VC(\mathcal{F})$, then under a given loss function $L : Y \times Y \rightarrow \mathbb{R}_+$, and for $f = \{f_k\}_{k=1}^{|\mathcal{D}|}$ where $f_k \in \mathcal{F}, \forall k \in [|\mathcal{D}|]$ with $f_k : T(\mathcal{X}) \rightarrow \mathbb{R}_+$ s.t. $\sup_{X \in \mathcal{X}} |f_k(T(X))| \leq M \in \mathbb{R}_+, \forall k \in [|\mathcal{D}|]$ derived from Lemma 4.2, consequently, $L(f_k(T(X), Y)) \leq \phi(M) \in \mathbb{R}_+, \forall k \in [|\mathcal{D}|], X \in \mathcal{X}, Y \in \mathcal{Y}$, where $\phi$ is some function of $M$, denote $k^* \leftarrow \arg\max_k |\hat{\mathcal{R}}^{LDP}(f_k) - \mathcal{R}^{LDP}(f_k)|$, if $n \geq \frac{8\ln(\frac{|\mathcal{D}|}{\delta})}{\min_k \mathbb{P}(S=k)}$ then with probability $1 - 2\delta$:

$$\hat{\mathcal{R}}^{LDP}(f) \leq \mathcal{R}(f^*) + K\sqrt{\frac{VC(\mathcal{F}) + \ln(\frac{\delta}{2})}{2n}} \frac{2C_1\phi(M)|\mathcal{D}|}{\mathbb{P}(S = k^*)}.$$

*Proof.* For better presentation, denote $X = T(X)$ and $\tilde{X} = \tilde{T}(X)$ in the proof.

**Step 1:** simplify the objective

Denote $\mathcal{R}(f_k)$ as the expected risk of $f_k$, and $\hat{\mathcal{R}}(f_k)$ as the empirical risk of $f_k$ that depends on the data set given, then we start with

$$\begin{aligned} &\mathbb{P}(|\hat{\mathcal{R}}^{LDP}(f) - \mathcal{R}(f)| > \epsilon) \\ =&\mathbb{P}(|\hat{\mathcal{R}}^{LDP}(f) + \mathcal{R}^{LDP}(f) - \mathcal{R}^{LDP}(f) - \mathcal{R}(f)| > \epsilon) \\ \leq&\mathbb{P}(|\hat{\mathcal{R}}^{LDP}(f) - \mathcal{R}^{LDP}(f)| + |\mathcal{R}^{LDP}(f) - \mathcal{R}(f)| > \epsilon) \\ \overset{(a)}{=}&\mathbb{P}(|\hat{\mathcal{R}}^{LDP}(f) - \mathcal{R}^{LDP}(f)| \geq \epsilon) \\ =&\mathbb{P}\Big(|\sum_{k=1}^{|\mathcal{D}|} \hat{\mathcal{R}}^{LDP}(f_k) - \sum_{k=1}^{D} \mathcal{R}^{LDP}(f_k)| > \epsilon\Big) \\ \leq&\mathbb{P}\Big(\sum_{k=1}^{D} |\hat{\mathcal{R}}^{LDP}(f_k) - \mathcal{R}^{LDP}(f_k)| > \epsilon\Big) \\ \overset{(b)}{\leq}&\mathbb{P}\Big(\max_k |\hat{\mathcal{R}}^{LDP}(f_k) - \mathcal{R}^{LDP}(f_k)| > \frac{\epsilon}{|\mathcal{D}|}\Big) \\ \overset{(c)}{=}&\mathbb{P}\Big(\Big|\sum_{j=1}^{|\mathcal{D}|} \hat{\boldsymbol{\Pi}}_{k^*j}^{-1} \frac{1}{n_j} \sum_{i:S_i=j} L\big(Y_i, f_{k^*}(T(X_i))\big) - \boldsymbol{\Pi}_{k^*\cdot}^{-1} \mathbb{E}_{Y,X|S}\Big[L(Y, f_{k^*}(X))\Big]\Big| > \frac{\epsilon}{|\mathcal{D}|}\Big), \end{aligned}$$

where (a) is obtained from the population equivalence of two losses from Lemma 4.2.

(b) is followed by for two events $A, B$, if $A$ implies $B$ then $P(A) < P(B)$, also denote that $k^* \leftarrow \arg\max_k |\hat{\mathcal{R}}^{LDP}(f_k) - \mathcal{R}^{LDP}(f_k)|$.

(c) is obtained by expanding the expression for $\hat{\mathcal{R}}^{LDP}(f_{k^*})$ and $\mathcal{R}^{LDP}(f_{k^*})$ respectively.

**Step 2:** concentration of the empirical risk under Risk-LDP

Denote $n_{yxs}^N = \sum_i \mathbf{1}(y_i = y, x_i = x, s_i = s)$, $\boldsymbol{Q}_{yxs} = \mathbb{P}(Y = y, X = x, S = j)$, and define the random variable $N_{yxs} = \{i \mid y_i = y, x_i = x, s_i = s\}$. We can deduce $n_s^N = \sum_{x \in X, y \in Y} \mathbf{1}(y_i = y, x_i = x, s_i = s)$. Then, we have $\mathbb{E}[\hat{\mathcal{R}}^{LDP}(f_{k^*}) \mid N_{YXS}] = \mathcal{R}^{LDP}(f_{k^*})$, where $N_{YXS}$ denotes all possible $N_{yxs}$. Using similar approach of Lemma 2 in Mozannar et al. (2020), we can write:

$$\mathbb{P}(\hat{\mathcal{R}}^{LDP}(f_k)^N - \mathcal{R}^{LDP}(f_{k^*}) > \frac{\epsilon}{|\mathcal{D}|})$$

$$\overset{(a)}{=} \sum_{N_{YXS}} \mathbb{P}\Big(\hat{\mathcal{R}}^{LDP}(f_{k^*})^N - \mathcal{R}^{LDP}(f_{k^*}) > \frac{\epsilon}{|\mathcal{D}|}\Big|N_{YXS}\Big) \cdot \mathbb{P}(N_{YXS})$$

$$\overset{(b)}{\leq} \mathbb{P}\Big(\bigcup_{x \in X, y \in Y, s \in S} \Big\{n_s^N < \frac{n \sum_{x \in X, y \in Y} \boldsymbol{Q}_{yxs}}{2}\Big\}\Big)$$

$$+ \sum_{\forall x, y, N_{yxs}: n_s^N \geq \frac{n \sum_{x \in X, y \in Y} \boldsymbol{Q}_{yxs}}{2}} \mathbb{P}\Big(\hat{\mathcal{R}}^{LDP}(f_{k^*})^N - \mathcal{R}^{LDP}(f_{k^*}) > \frac{\epsilon}{|\mathcal{D}|}\Big|N_{YXS}\Big) \cdot \mathbb{P}(N_{YXS})$$

$$\overset{(c)}{\leq} |\mathcal{D}| \exp\Big\{ - \frac{\min_s n \sum_{x \in X, y \in Y} \boldsymbol{Q}_{yxs}}{8}\Big\}$$

$$+ \sum_{\forall x, y, N_{yxs}: n_s^N \geq \frac{n \sum_{x \in X, y \in Y} \boldsymbol{Q}_{yxs}}{2}} \mathbb{P}\Big(\hat{\mathcal{R}}^{LDP}(f_{k^*})^N - \mathcal{R}^{LDP}(f_{k^*}) > \frac{\epsilon}{|\mathcal{D}|}\Big|N_{YXS}\Big) \cdot \mathbb{P}(N_{YXS}),$$

where (a) follows by conditioning over all $2^n |X|^n |\mathcal{D}|^n$ possible configurations of $N_{yxs} \subset [n]$. (b) is obtained by splitting the configurations where $\forall x, y, N_{yxs} : n_s^N \geq \frac{n \sum_{x \in X, y \in Y} \boldsymbol{Q}_{yxs}}{2}$ and the complement of the event and upper bound the complement of the event by the probability that $\exists s$ s.t. $n_s^N < \frac{n \sum_{x \in X, y \in Y} \boldsymbol{Q}_{yxs}}{2}$. (c) is obtained by the union bound and we know $n_s^N \sim$ Binomial$\big(n, \sum_{x \in X, y \in Y} \boldsymbol{Q}_{yxs}\big)$ and apply the Chernoff bound on $n_{yxj}^N$.

Now, we will apply the McDiarmid Inequality McDiarmid (1989). Let $X^n = (X_1, \ldots, X_n) \in X^n$ be n independent random variables and let $g : X^n \to \mathbb{R}$, if there exists constants $c_1, \ldots, c_n$ s.t.

$$\sup_{x_1, \ldots, x_i, x_i', \ldots, x_n} |g(x_1, \ldots, x_i, \ldots, x_n) - g(x_1, \ldots, x_i', \ldots, x_n)| \leq c_i, i = 1, \ldots, n,$$

then $\forall \epsilon > 0$:

$$\mathbb{P}(g(x_1, \ldots, x_i, \ldots, x_n) - \mathbb{E}[g(x_1, \ldots, x_i, \ldots, x_n)]) \leq 2\exp\Big(\frac{2\epsilon^2}{\sum_{i=1}^n c_i^2}\Big).$$

Since by conditioning on $N_{YXS}$, then for $\hat{\mathcal{R}}^{LDP}(f_{k^*})$, everything else is now deterministic except for $f_{k^*}$, in other words, by conditioning on $N_{YXS}$, the value of $\hat{\mathcal{R}}^{LDP}(f_{k^*})$ only depends on $f_{k^*}$. Then, for two datasets $N, N'$ where they only differ by one value of $f_{k^*}(X_i)$, we try to bound how much $f_{k^*}$ can change.

Recall from Lemma 4.2, we computed the entries of $\boldsymbol{\Pi}^{-1}$ takes the following form:

$$\begin{cases} \boldsymbol{\Pi}_{i,i}^{-1} = \frac{\pi + |\mathcal{D}| - 2}{|\mathcal{D}|\pi - 1} \frac{\pi p_i + \sum_{d'' \setminus i} \bar{\pi} p_{d''}}{p_i}, \text{ for } i \in D \\ \boldsymbol{\Pi}_{i,j}^{-1} = \frac{\pi - 1}{|\mathcal{D}|\pi - 1} \frac{\bar{\pi} p_i + \sum_{d'' \setminus i} \pi p_{d''}}{p_i}, \text{ for } i, j \in D \text{ s.t.}, i \neq j \end{cases}.$$

For simplicity, let $C_1 = \frac{\pi + |\mathcal{D}| - 2}{|\mathcal{D}|\pi - 1}$, $C_2 = \frac{\pi - 1}{|\mathcal{D}|\pi - 1}$, since we do not have access to $D$, therefore we can not directly observe $p_d$, hence we write $\mathbf{\Pi}^{-1}$ in terms of $\boldsymbol{P}_s$ where $\boldsymbol{P}_s = \mathbb{P}(S)$:

$$
\begin{cases}
\mathbf{\Pi}^{-1}_{i,i} = C_1 \frac{\pi \boldsymbol{T}^{-1}_{i\cdot} \boldsymbol{P}_s + \sum_{l\setminus i} \bar{\pi} \boldsymbol{T}^{-1}_{l\cdot} \boldsymbol{P}_s}{\boldsymbol{T}^{-1}_{i\cdot} \boldsymbol{P}_s}, \text{ for } i \in D \\
\mathbf{\Pi}^{-1}_{i,j} = C_2 \frac{\bar{\pi} \boldsymbol{T}^{-1}_{i\cdot} \boldsymbol{P}_s + \sum_{l\setminus i} \pi \boldsymbol{T}^{-1}_{l\cdot} \boldsymbol{P}_s}{\boldsymbol{T}^{-1}_{i\cdot} \boldsymbol{P}_s}, \text{ for } i,j \in D \text{ s.t.,} i \neq j
\end{cases} ,
$$

we also computed $\boldsymbol{T}^{-1}$ as:

$$
\begin{cases}
\boldsymbol{T}^{-1}_{i,i} = \frac{\pi + |\mathcal{D}| - 2}{|\mathcal{D}|\pi - 1}, \text{ for } i \in D \\
\boldsymbol{T}^{-1}_{i,j} = \frac{\pi - 1}{|\mathcal{D}|\pi - 1}, \text{ for } i,j \in D \text{ s.t.,} i \neq j
\end{cases} ,
$$

then we have

$$
\sup_{N,N'} |\hat{\mathcal{R}}^{LDP}(f_{k^*})^N - \hat{\mathcal{R}}^{LDP}(f_{k^*})^{N'}|
$$

$$
= \Bigg| C_1 \frac{\pi \boldsymbol{T}^{-1}_{k^*\cdot} \boldsymbol{P}^N_s + \sum_{l\setminus k} \bar{\pi} \boldsymbol{T}^{-1}_{l\cdot} \boldsymbol{P}^N_s}{\boldsymbol{T}^{-1}_{k^*\cdot} \boldsymbol{P}^N_s} \hat{\mathcal{R}}^{LDP}(f_{k^*})^N + \sum_{j\setminus k} C_2 \frac{\bar{\pi} \boldsymbol{T}^{-1}_{k^*\cdot} \boldsymbol{P}^N_s + \sum_{l\setminus k} \pi \boldsymbol{T}^{-1}_{l\cdot} \boldsymbol{P}^N_s}{\boldsymbol{T}^{-1}_{k^*\cdot} \boldsymbol{P}^N_s} \hat{\mathcal{R}}^{LDP}(f_{k^*})^N
$$

$$
- C_1 \frac{\pi \boldsymbol{T}^{-1}_{k^*\cdot} \boldsymbol{P}^N_s + \sum_{l\setminus k} \bar{\pi} \boldsymbol{T}^{-1}_{l\cdot} \boldsymbol{P}^N_s}{\boldsymbol{T}^{-1}_{k^*\cdot} \boldsymbol{P}^N_s} \hat{\mathcal{R}}^{LDP}(f_{k^*})^{N'} + \sum_{j\setminus k} C_2 \frac{\bar{\pi} \boldsymbol{T}^{-1}_{k^*\cdot} \boldsymbol{P}^N_s + \sum_{l\setminus k} \pi \boldsymbol{T}^{-1}_{l\cdot} \boldsymbol{P}^N_s}{\boldsymbol{T}^{-1}_{k^*\cdot} \boldsymbol{P}^N_s} \hat{\mathcal{R}}^{LDP}(f_{k^*})^{N'} \Bigg|
$$

$$
= \Bigg| C_1 \frac{\pi \boldsymbol{T}^{-1}_{k^*\cdot} \boldsymbol{P}^N_s + \sum_{l\setminus k} \bar{\pi} \boldsymbol{T}^{-1}_{l\cdot} \boldsymbol{P}^N_s}{\boldsymbol{T}^{-1}_{k^*\cdot} \boldsymbol{P}^N_s} \bigg( \frac{\sum_{i\in N, x\in X, y\in Y, S=k} L(y_i, f_{k^*}(x_i))}{n_{k^*}} - \frac{\sum_{i\in N', x\in X, y\in Y, S=k} L(y_i, f_{k^*}(x_i))}{n_{k^*}} \bigg)
$$

$$
+ \sum_{j\setminus k} C_2 \frac{\bar{\pi} \boldsymbol{T}^{-1}_{k^*\cdot} \boldsymbol{P}^N_s + \sum_{l\setminus k} \pi \boldsymbol{T}^{-1}_{l\cdot} \boldsymbol{P}^N_s}{\boldsymbol{T}^{-1}_{k^*\cdot} \boldsymbol{P}^N_s} \bigg( \frac{\sum_{i\in N, x\in X, y\in Y, S=j} L(y_i, f_{k^*}(x_i))}{n_{k^*}} - \frac{\sum_{i\in N', x\in X, y\in Y, S=j} L(y_i, f_{k^*}(x_i))}{n_{k^*}} \bigg) \Bigg|
$$

$$
\overset{(a)}{\leq} \Bigg| C_1 \frac{\pi \max_{m\in[|\mathcal{D}|]} \boldsymbol{T}^{-1}_{m\cdot} \boldsymbol{P}^N_s + (|\mathcal{D}| - 1)\bar{\pi} \max_{m\in[|\mathcal{D}|]} \boldsymbol{T}^{-1}_{m\cdot} \boldsymbol{P}^N_s}{\boldsymbol{T}^{-1}_{k^*\cdot} \boldsymbol{P}^N_s} \cdot \frac{\phi(M)}{n_{k^*}} \Bigg|
$$

$$
= \Bigg| C_1 \big( \pi + \bar{\pi}(|\mathcal{D}| - 1) \big) \cdot \frac{\phi(M)}{n_{k^*}} \Bigg|
$$

$$
\overset{(b)}{=} \Bigg| \frac{C_1 \phi(M)}{n_{k^*}} \Bigg| ,
$$

where (a) is obtained by $C_2 \leq 0, \forall \pi \in (\frac{1}{|\mathcal{D}|}, 1]$. (b) is followed by the fact that $\pi + \bar{\pi}(|\mathcal{D}| - 1) = 1$.

Now, we are ready to apply the McDiarmid Inequality:

$$
\sum_{\forall x,y,N_{yxs} : n^N_s \geq \frac{n \sum_{x\in X, y\in Y} \boldsymbol{Q}_{yxs}}{2}} \mathbb{P}\Big( \hat{\mathcal{R}}^{LDP}(f_{k^*})^N - \mathcal{R}^{LDP}(f_{k^*}) > \frac{\epsilon}{|\mathcal{D}|} \Big| N_{YXS} \Big) \cdot \mathbb{P}(N_{YXS})
$$

$$
\leq \sum_{\forall x,y,N_{yxs} : n^N_s \geq \frac{n \sum_{x\in X, y\in Y} \boldsymbol{Q}_{yxs}}{2}} 2 \exp \left\{ - \frac{\frac{2\epsilon^2}{|\mathcal{D}|^2}}{n \cdot \big( \frac{C_1 \phi(M)}{n_{k^*}} \big)^2} \right\} \cdot \mathbb{P}(N_{YXS})
$$

$$
\overset{(a)}{\leq} 2 \exp \left\{ - 2n\epsilon^2 \Big( \frac{\mathbb{P}(S=k)}{2C_1 \phi(M)|\mathcal{D}|} \Big)^2 \right\},
$$

where (a) is obtained since when $n_{k^*} = \frac{n \sum_{x\in X, y\in Y} \boldsymbol{Q}_{yxk}}{2} = \frac{n\mathbb{P}(S=k)}{2}$, the quantity is maximized.

Now, we have:

$$
\mathbb{P}(|\hat{\mathcal{R}}^{LDP}(f) - \mathcal{R}(f)| > \epsilon) \leq |\mathcal{D}| \exp \left\{ - \frac{\min_k \mathbb{P}(S=k)}{8} \right\} + 2 \exp \left\{ - 2n\epsilon^2 \Big( \frac{\mathbb{P}(S=k^*)}{2C_1 \phi(M)|\mathcal{D}|} \Big)^2 \right\},
$$

solve for $\delta$, we now have, for any $\delta \in (0, \frac{1}{2})$, $\epsilon \geq \sqrt{\frac{\ln(\frac{\delta}{2})}{2n}} \frac{2C_1\phi(M)|\mathcal{D}|}{\mathbb{P}(S=k^*)}$, if $n \geq \frac{8\ln(\frac{|\mathcal{D}|}{\delta})}{\min_k \mathbb{P}(S=k)}$, then

$$\mathbb{P}(|\hat{\mathcal{R}}^{LDP}(f) - \mathcal{R}(f)| > \epsilon) \leq 2\delta$$

**Step 3:** Obtain the final result

Recall that one can easily show

$$\hat{\mathcal{R}}^{LDP}(f) - \mathcal{R}(f^*) \leq 2\sup_{f \in \mathcal{F}}|\mathcal{R}^{LDP}(f) - \hat{\mathcal{R}}^{LDP}(f)|,$$

but we have already established similar results for one single hypothesis in **Step 2**. Therefore, what remains is to extend the previous result that bounds the generalization error between any single hypothesis and the optimal hypothesis in the entire hypothesis class. And this can be done easily by introducing the VC-dimension of the hypothesis $\mathcal{F}$. Denote the VC-dimension of our hypothesis class $\mathcal{F}$ as $VC(\mathcal{F})$, then with some constant $K$ and for any $\delta \in (0, \frac{1}{2})$, if $n \geq \frac{8\ln(\frac{|\mathcal{D}|}{\delta})}{\min_k \mathbb{P}(S=k)}$, we have:

$$\hat{\mathcal{R}}^{LDP}(f) \leq \mathcal{R}(f^*) + K\sqrt{\frac{VC(\mathcal{F}) + \ln(\frac{\delta}{2})}{2n}} \frac{2C_1\phi(M)|\mathcal{D}|}{\mathbb{P}(S=k^*)}.$$

This completes the proof. $\qquad\square$

### G.4 THEOREM 4.5 PROOF

**Theorem 4.5** For any $\delta \in (0, \frac{1}{3})$, $C_1 = \frac{\pi + |\mathcal{D}| - 2}{|\mathcal{D}|\pi - 1} > 0$, $\hat{C}_1 = \frac{1}{n_1}\sum_{k=1}^{n_1} \hat{C}_{1,k}$, where $\hat{C}_{1,k}$ is defined in Lemma 4.4, denote $VC(\mathcal{F})$ as the VC-dimension of the hypothesis class $\mathcal{F}$, and $K$ be some constant that depends on $VC(\mathcal{F})$, if Assumption A (4.3),B (4.3), and Lemma 4.4 hold, given a loss function $L : Y \times Y \to \mathbb{R}_+$, $M_g + \frac{C_1 + \theta}{\ln 2} > \tilde{\epsilon} > \theta$, and for $f = \{f_k\}_{k=1}^{|\mathcal{D}|}$ where $f_k \in \mathcal{F}, \forall k \in [|\mathcal{D}|]$ with $f_k : T(\mathcal{X}) \to \mathbb{R}_+$ s.t. $\sup_{X \in \mathcal{X}}|f_k(T(X))| \leq M \in \mathbb{R}_+, \forall k \in [|\mathcal{D}|]$ derived from Lemma 4.2, consequently, $L(f_k(T(X), Y)) \leq \phi(M) \in \mathbb{R}_+, \forall k \in [|\mathcal{D}|], X \in \mathcal{X}, Y \in \mathcal{Y}$, where $\phi$ is some function of $M$, denote $k^* \leftarrow \arg\max_k|\hat{\mathcal{R}}^{LDP}(f_k) - \mathcal{R}^{LDP}(f_k)|$, if $n \geq \frac{8\ln(\frac{|\mathcal{D}|}{\delta})}{\min_k \mathbb{P}(S=k)}$, $n_1 \geq \frac{1}{c(\tilde{\epsilon}-\theta)^2}(M_g + \frac{C_1+\theta}{\ln 2})^2\ln(\frac{2}{\delta})$ where $c$ is an absolute constant, then with probability $1 - 3\delta$:

$$\hat{\mathcal{R}}^{LDP}(f) \leq \mathcal{R}(f^*) + K\sqrt{\frac{VC(\mathcal{F}) + \ln(\frac{\delta}{2})}{2n}} \frac{2(C_1 + \tilde{\epsilon})\phi(M)|\mathcal{D}|}{\mathbb{P}(S=k^*)}.$$

*Proof.* We will first introduce some preliminaries that will be used in the proof. We will first introduce how we obtain $\hat{C}_1$ and then state the assumptions used for the proof.

**Step 1: Grouping:** Given the observed data $\{T(X_i), S_i\}_{i=1}^n$, we evenly divide them into $n_1$ groups, with $m = \frac{n}{n_1}$ samples each.

**Step 2: Estimating within groups:** for any $k \in [n_1]$, within every group $\{T(X_{k,j}), S_{k,j}\}_{j=1}^m$, we can derive an $m$-dimension vector $\boldsymbol{\eta}_{s,k}(\tilde{T}(X_{k,\cdot})^*) = (\hat{\mathbb{P}}_k(S = j^*|\tilde{T}(X_{k,1})), \ldots, \hat{\mathbb{P}}_k(S = j^*|\tilde{T}(X_{k,m})))$ and $\hat{\pi}_k = \|\boldsymbol{\eta}_{s,k}(T(X)^*)\|_\infty$, which is defined in Lemma 4.4. Then, applying a straight-forward plug in $\hat{C}_{1,k} = \frac{\hat{\pi}_k + |\mathcal{D}| - 2}{|\mathcal{D}|\hat{\pi}_k - 1}$.

**Step 3:Averaging:** Finally, our estimator for $C_1$, denoted by $\hat{C}_1 = \frac{1}{n_1}\sum_{k=1}^{n_1} \hat{C}_{1,k}$, can be derived by averaging $\hat{C}_{1,k}, k \in [n_1]$.

Next, we state two assumptions that we used to derive the generalization error bound for Risk-LDP (Eq. (7)) when the noise rate is estimated from the data.

**Assumption A:** (Sub-exponentiality) For all $k \in [n_1]$, define $\hat{g}_k(\tilde{T}(X)) = \hat{\mathbb{P}}_k(S = j^*|\tilde{T}(X))$ There exists a constant $M_g > 0$, such that $\|\hat{C}_{1,k}\|_{\psi_1} = \|\min_{i \in [m]} \frac{\hat{g}_k(\tilde{T}(X_{k,i})) + |\mathcal{D}| - 2}{|\mathcal{D}|\hat{g}_k(\tilde{T}(X_{k,i})) - 1}\|_{\psi_1} \leq M_g$ for all

$k \in [n_1]$, where $\| \cdot \|_{\psi_1}$ is the sub-exponential norm:

$$\|X\|_{\psi_1} = \inf\{t > 0 | \mathbb{E}[e^{X/t}] \leq 2\}.$$

**Assumption B:** (Nearly Unbiasedness) For all $k \in [n_1]$, $\hat{C}_{1,k}$ is a 'nearly' unbiased estimator of $C_1$, namely $\left|\mathbb{E}[\hat{C}_{1,k}] - C_1\right| < \theta$ for all $k \in [m]$, where $\theta > 0$.

Now, we begin the proof.

**First**, we will prove a concentration inequality with regard to $\hat{C}_1$ and $C_1$.

Since for any constant $L$, we have

$$\begin{aligned}
\|L\|_{\psi_1} &= \inf\{t > 0 | \mathbb{E}[e^{|L|/t}] \leq 2\} \\
&= \inf\{t > 0 | e^{|L|/t} \leq 2\} \\
&= \frac{|L|}{\ln 2},
\end{aligned}$$

and $\| \cdot \|_{\psi_1}$ is a norm, we can conclude that the standardized statistic $\tilde{C}_{1,k} = \hat{C}_{1,k} - \mathbb{E}[\hat{C}_{1,k}]$ is also sub-exponential:

$$\begin{aligned}
\|\tilde{C}_{1,k}\|_{\psi_1} &\leq \|\hat{C}_{1,k}\|_{\psi_1} + \|\mathbb{E}[\hat{C}_{1,k}]\|_{\psi_1} \\
&\leq M_q + \frac{|\mathbb{E}[\hat{C}_{1,k}]|}{\ln 2} \\
&\overset{(a)}{=} M_q + \frac{C_1 + \theta}{\ln 2},
\end{aligned}$$

where (a) is obtained by Assumption B.

Among different groups, the data are mutually independent, then we know that $\{\tilde{C}_{1,k}\}_{k=1}^{n_1}$ are independent random variables with mean 0.

Therefore, we can apply Bernstein inequality(R.Vershynin (2018)):

$$\mathbb{P}\left(\left|\frac{1}{n_1}\sum_{k=1}^{n_1}\tilde{C}_{1,k}\right| > \tilde{\epsilon} + \theta\right) \leq 2\exp\left[-c\min\left(\frac{(\tilde{\epsilon}+\theta)^2}{(M_g + C_1/\ln 2)^2}, \frac{\tilde{\epsilon}+\theta}{M_g + C_1/\ln 2}\right)n_1\right],$$

where $c > 0$ is an absolute constant.

Since we have $M_g + \frac{C_1+\theta}{\ln 2} > \tilde{\epsilon} > \theta$, which implies $\frac{\tilde{\epsilon}}{M_g + C_1/\ln 2} < 1$, we can transform the inequality above into

$$\begin{aligned}
\mathbb{P}\left(\left|\hat{C}_1 - C_1\right| > \tilde{\epsilon}\right) &= \mathbb{P}\left(\left|\frac{1}{n_1}\sum_{k=1}^{n_1}\tilde{C}_{1,k}\right| > \tilde{\epsilon} - \theta\right) \\
&\leq 2\exp\left[-c\frac{(\tilde{\epsilon}-\theta)^2}{(M_g + (C_1+\theta)/\ln 2)^2}n_1\right] \\
&\overset{(a)}{\leq} \delta,
\end{aligned}$$

where (a) is obtained by $n_1 \geq \frac{1}{c(\tilde{\epsilon}-\theta)^2}(M_g + \frac{C_1+\theta}{\ln 2})^2 \ln(\frac{2}{\delta})$.

**Second**, we can apply Theorem 4.3 to the case when using $\hat{\pi}$ instead of $\pi$. Therefore, by the end of **Step 2** in the proof of Theorem 4.3, we will derive the following conclusion:

For any $\delta \in (0, \frac{1}{3})$, $\epsilon \geq \sqrt{\frac{\ln(\frac{\delta}{2})}{2n}}\frac{2\hat{C}_1\phi(M)|\mathcal{D}|}{\mathbb{P}(S=k^*)}$, if $n \geq \frac{8\ln(\frac{|\mathcal{D}|}{\delta})}{\min_k \mathbb{P}(S=k)}$, then

$$\mathbb{P}\left(|\hat{\mathcal{R}}^{LDP}(f) - \mathcal{R}(f)| > \epsilon\right) \leq 2\delta.$$

**Third**, Assume the events

$$A_1 = \left\{ \left| \hat{C}_1 - C_1 \right| \le \tilde{\epsilon} \right\},$$

$$A_2 = \left\{ |\hat{\mathcal{R}}^{LDP}(f) - \mathcal{R}(f)| \le \epsilon, \epsilon \ge \sqrt{\frac{\ln\left(\frac{\delta}{2}\right)}{2n} \frac{2\hat{C}_1\phi(M)|\mathcal{D}|}{\mathbb{P}(S=k^*)}}, n \ge \frac{8\ln\left(\frac{|\mathcal{D}|}{\delta}\right)}{\min_k \mathbb{P}(S=k)} \right\},$$

$$A_3 = \left\{ |\hat{\mathcal{R}}^{LDP}(f) - \mathcal{R}(f)| \le \epsilon, \epsilon \ge \sqrt{\frac{\ln\left(\frac{\delta}{2}\right)}{2n} \frac{2(C_1+\tilde{\epsilon})\phi(M)|\mathcal{D}|}{\mathbb{P}(S=k^*)}}, n \ge \frac{8\ln\left(\frac{|\mathcal{D}|}{\delta}\right)}{\min_k \mathbb{P}(S=k)} \right\},$$

then we have $A_1 \cap A_2 \subseteq A_3$.

From the **First** part and **Second** part of the proof, we have $\mathbb{P}(A_1^C) \le \delta, \mathbb{P}(A_2^C) \le 2\delta$, then

$$\mathbb{P}(A_3) \ge \mathbb{P}(A_1 \cap A_2) \ge 1 - \mathbb{P}(A_1^C) - \mathbb{P}(A_2^C) \ge 1 - 3\delta,$$

which is equivalent to the following statement: For any $\delta \in (0, \frac{1}{3}), \epsilon \ge \sqrt{\frac{\ln\left(\frac{\delta}{2}\right)}{2n} \frac{(2(C_1+\tilde{\epsilon})\phi(M)|\mathcal{D}|}{\mathbb{P}(S=k^*)}}$, if $n \ge \frac{8\ln\left(\frac{|\mathcal{D}|}{\delta}\right)}{\min_k \mathbb{P}(S=k)}$, then

$$\mathbb{P}\left(|\hat{\mathcal{R}}^{LDP}(f) - \mathcal{R}(f)| > \epsilon\right) \le 3\delta.$$

**Finally**, similar to **Step 3** in the proof of Theorem 4.3, recall that one can easily show

$$\hat{\mathcal{R}}^{LDP}(f) - \mathcal{R}(f^*) \le 2 \sup_{f \in \mathcal{F}} \left| \mathcal{R}^{LDP}(f) - \hat{\mathcal{R}}^{LDP}(f) \right|,$$

but we have already established similar results for one single hypothesis in **Step 2**. Therefore, what remains is to extend the previous result that bounds the generalization error between any single hypothesis and the optimal hypothesis in the entire hypothesis class. And this can be done easily by introducing the VC-dimension of the hypothesis $\mathcal{F}$. Denote the VC-dimension of our hypothesis class $\mathcal{F}$ as $VC(\mathcal{F})$, then with some constant $K$ and for any $\delta \in (0, \frac{1}{3})$, if $n \ge \frac{8\ln\left(\frac{|\mathcal{D}|}{\delta}\right)}{\min_k \mathbb{P}(S=k)}$, then with probability $1 - 3\delta$ we have:

$$\hat{\mathcal{R}}^{LDP}(f) \le \mathcal{R}(f^*) + K\sqrt{\frac{VC(\mathcal{F}) + \ln\left(\frac{\delta}{2}\right)}{2n} \frac{2(C_1+\tilde{\epsilon})\phi(M)|\mathcal{D}|}{\mathbb{P}(S=k^*)}}.$$

This completes the proof. $\qquad\square$

