# OpenReview forum: "Discrimination-free Pricing with Privatized Sensitive Attributes"
_ICLR.cc/2024/Conference — Submitted to ICLR 2024_

### Official Review · Reviewer_rqKi · 2023-10-30

**Soundness:** 2 fair
**Presentation:** 3 good
**Contribution:** 2 fair
**Rating:** 6
**Confidence:** 2

**Summary:**

Due to difficulty learning fair insurance prices because of inaccessible/privatized sensitive attribute data, the authors propose a multi-party training framework to achieve discrimination-free insurance pricing. In their proposed model, the insurer has access to all data except the sensitive attributes and the third party uses the transformed data from the insurer plus the sensitive information to make fair pricing predictions. The authors test their method on the Adult income dataset and compare accuracy for varied values of privacy budget.

**Strengths:**

- I like the problem the authors investigate. The authors tackle a real challenge faced during fair predictive decision-making.

- The write-up is precise and consistent, and the ideas are well presented.

- I like that authors theoretically and empirically investigated cases of known and unknown pi and showed results for varied privacy budgets.

**Weaknesses:**

While I think the authors did a great job laying down the proposed model, I observed some shortcomings that influenced my score.
Below are the observed weaknesses (and respective suggestions) and some questions.

- Specific versus general models. Although the authors mention that the biggest strength of their work (especially in comparison to previous works -relatedworks) is their model working under any given loss function, in their theory and empirical work, the focus is on logistic regression. The authors (reasonably) defend this choice as a tradeoff between transparency and complexity, which makes it hard to appreciate author contributions.

- Since noise levels have a significant effect on risk-LDP, I am curious about the effect of overestimating and underestimating noise on fairness and the impact on different sensitive groups, especially in the case of unevenly distributed groups.

- Several challenges are associated with a multi-party framework, for example, information leakage, computation overhead, etc..  I am curious how the proposed method would be comparatively better than those settings where fairness is computed on (single-party) fully differentially private data (X,D) or where causal inference (and other methods) is used to perform fairness in the absence of sensitive attributes.

- Experimental setup and results. There are other (single-party) fair decision-making with private data methods and noisy sensitive attributes that the authors could have compared their work with. Although authors show different error rates with varied privacy budgets, it would have been informative to see how the method compares to other (similar) methods. Additionally, authors say they couldn't find insurance-like data, but there are at least 50 insurance datasets on Data World (and other platforms).

**Questions:**

Although generally, I think the authors did a great job with outlining the problem and proposed solution, I found a couple of shortcomings (questions) raised in the weakness section that influenced my score.

---

> ### Author Response · Authors · 2023-11-17
> **Respond to Reviewer rqKi**
>
> We thank the reviewer for appreciating the preciseness and consistency of our presentation. We also thank the reviewer for acknowledging the practical potential of our work in tackling the real challenges faced by fair insurance pricing. Further, we value the constructive feedback that the reviewer provided as it contributed to the improvement of our paper. In the following, we will carefully respond to each of the concerns and suggestions that the reviewer raised.
>
> $\textbf{Q1:}$ "pecific versus general models. Although the authors ......  transparency and complexity, which makes it hard to appreciate author contributions."
>
> $\textbf{A1:}$ We agree with the reviewer's opinion on presenting empirical experiments on losses other than logistic (Cross-entropy) loss to be consistent with what the derived theoretical results claim. Further, an insurance-related data set shall be used to conduct empirical experiments. Therefore, we added a regression task on healthcare insurance premium pricing (thank the reviewer for directing us to the data source). As we should expect, the results from this empirical experiment using MSE loss are again in support of our derived theoretical results. Please see Section 5 of the revised paper for a detailed discussion.

---

> > ### Author Response · Authors · 2023-11-17
> > **Respond to Reviewer rqKi**
> >
> > $\textbf{Q2:}$ "Since noise levels have a significant effect on risk-LDP, I am curious about the effect of overestimating and underestimating noise on fairness and the impact on different sensitive groups, especially in the case of unevenly distributed groups."
> >
> > $\textbf{A2:}$ To answer the reviewer's question on the impact of noise rate estimation error on Risk-LDP, we added an empirical study. A high-level result of the study is that when $\pi$ is far away from $\frac{1}{|\mathcal{D}|}$, Risk-LDP is robust against estimation error even if the error is large (regardless of overestimation or underestimation). However, when $\pi$ gets close to $\frac{1}{|\mathcal{D}|}$, Risk-LDP suffers from both underestimation (more severe) and overestimation even if the error is small. Please see Section 5.3, Appendix E for figures and a more detailed discussion.

---

> > > ### Author Response · Authors · 2023-11-17
> > > **Respond to Reviewer rqKi**
> > >
> > > $\textbf{Q3:}$ "Several challenges are associated ...... to perform fairness in the absence of sensitive attributes."
> > >
> > > $\textbf{A3:}$  All algorithms that involve multiple parties suffer from the shortcomings the reviewer listed (i.e. information leakage, computation overhead, etc..) But, one merit of our framework is that it is not limited to the multi-party setting, multi-party training process is introduced as an alternative when the insurer cannot directly access all the data needed to train a discrimination-free pricing model. We mentioned two settings where this multi-party training process will be reduced to a regular single-party training process.:
> > >
> > > 1) The insurer is able to obtain the privatized sensitive attributes from a third party and apply the proposed algorithm directly.
> > > 2) The insurer collects information on both non-sensitive and sensitive attributes and sends this information to a third-party vendor to execute the pricing algorithm.
> > >
> > > Therefore, even if we assume that the insurer has direct access to all the data needed, meaning that our framework is reduced to single-party training, we do not think our method is directly comparable to methods that compute fair predictions satisfying the fairness notion defined in the conventional algorithmic fairness literature because we are aiming at achieving different fairness notions.
> > >
> > > Since discrimination-free pricing in insurance is a new topic, there are only a few papers that give insights from the causal inference perspective, however, all of them require direct access to the true sensitive attributes, and no discrimination-free pricing algorithms that consider sensitive attributes privatization has been developed in the discrimination-free insurance pricing literature to our best knowledge.

---

> > > > ### Author Response · Authors · 2023-11-17
> > > > **Respond to Reviewer rqKi**
> > > >
> > > > $\textbf{Q4:}$ "Experimental setup and results. There are other (single-party) fair ......  datasets on Data World (and other platforms).
> > > >
> > > > $\textbf{A4:}$ As mentioned in some parts of $\textbf{A1, A3}$. We added another empirical experiment to test the performance of our proposed algorithm in a regression setting and found the results also in support of our theoretical guarantees (please see Section 5 for a more detailed discussion). We'd like to point out that we are the first to propose algorithms that train discrimination-free models that rely solely on privatized sensitive attributes in the actuarial science literature. Therefore, we might not be able to find an appropriate paper for a direct comparison of the results from empirical experiments.

---

> ### Comment · Reviewer_rqKi · 2023-11-19
>
> First, I would like to thank the authors for addressing the concerns we raised and making changes to the paper. I have a few follow-up questions and comments;
> - When I examined the added experiments on over(under)estimation of added noise, I was unable to determine whether the groups were evenly or unevenly distributed. Also, can the authors comment on the real-world applicability of assumption B and alternatives when it doesn't hold?
> - The added sentence, ``Hence ours is the first work that does not rely on direct access to true sensitive attributes to train discrimination-free models.`` might not truly capture the state of fair ML.
>  - ``First, the insurer can obtain the privatized sensitive attributes from a third party and apply the proposed algorithm directly. Second, the insurer collects information on both non-sensitive and sensitive attributes and sends this information to a third-party vendor to execute the pricing algorithm. Furthermore, the proposed algorithm is readily applicable when non-sensitive attributes originate from third-party sources.``   From this paragraph, it looks like both the insurer (first) and the third party can (second) have access to the algorithm to do fair pricing, and in the second scenario, I am unsure of the need to send information if they can do pricing themselves. Additionally,  for the first case, I think this would contradict with the outlined procedure in 4.1. Would that mean the insurer doesn't do T(X), and h is performed on X? Lastly, since the insurer performing h seems cheaper, less noisy, probably fairer, etc., than sending to TTP, even in the absence of sensitive attributes, I am still unsure of how the multi-party setting would be an incentive to the insurer.

---

> ### Author Response · Authors · 2023-11-20
> **Respond to Reviewer rqKi**
>
> We'd like to begin by expressing our heartfelt gratitude for the time and effort you spent reviewing our revision. Your insights and feedback are very valuable in improving the quality of our work. We are more than willing to address your further concerns and questions based on our first revision. In the following, we will carefully respond to each of the concerns and suggestions that the reviewer raised.
>
> $\textbf{Q1:}$ "When I examined the added experiments ...... assumption B and alternatives when it doesn't hold?"
>
> $\textbf{A1 Part 1:}$ We apologize for the impreciseness in our first revision. In the second revision, we manually created two subsets where the ratio of the sensitive attributes (gender) are $\frac{F}{M} = \frac{4}{1}$ and $\frac{F}{M} = \frac{2}{1}$ respectively and we compared the results with the evenly distributed scenario. The results on a high level is that given a transformation, the convergence behavior of Risk-LDP is not very much impacted by the imbalance distribution, however, as the distribution of sensitive attributes becomes more imbalanced, Risk-LDP tends to give a higher loss. Please see Section 5 and Appendix E for a more detailed discussion.
>
> $\textbf{A1 Part 2:}$ Your question on the alternative of assumption B has inspired us to come up with a modified Theorem 4.5 with a relaxed assumption on the unbiasedness of $\hat{C}_{1,k}$. Formally, we do not require exact unbiasedness for Theorem 4.5 to hold, some perturbation on the unbiasedness is allowed. Please see Section 4.3, Appendix C.3, and Appendix G.4 for a more detailed discussion.

---

> > ### Author Response · Authors · 2023-11-20
> > **Respond to Reviewer rqKi**
> >
> > $\textbf{Q2:}$ "The added sentence, Hence ...... the state of fair ML."
> >
> > $\textbf{A2:}$ Yes, we agree that the added sentence in the first revision is ambiguous in a way that it mixes the algorithmic fairness literature and actuarial science literature and it is certainly not true in the algorithmic fairness literature. However, after a revisit to existing work on fair pricing in actuarial science literature. We are confident that "ours is the first work that considers the real-world challenges in training discrimination-free insurance pricing models in the actuarial science literature, that is the need of direct access to true sensitive attributes is relaxed to only a noisy version of true sensitive attributes." We have also made this correction in our second revision.

---

> > > ### Author Response · Authors · 2023-11-20
> > > **Respond to Reviewer rqKi**
> > >
> > > $\textbf{Q3:}$ "First, the insurer can obtain the privatized sensitive attributes ...... multi-party setting would be an incentive to the insurer."
> > >
> > > $\textbf{A3:}$ Let us elaborate on this with more details.
> > >
> > > Note: Before we formally address the reviewer's concerns and questions, please allow us to include some more industry background based on our past experience to better support the value and the need of solving this problem motivated by real-world industry needs.
> > >
> > > Case 1: "First, the insurer can obtain the privatized sensitive attributes from a third party and apply the proposed algorithm directly." This often happens in large-size (tier-1) insurance companies (i.e. State Farm, Allstate, etc), where they can afford a large research and pricing department and support with the computation power needed. Therefore, most of the time, pricing is done within the firm.
> > >
> > > However, there are times such as in preparation for new market development, outside parties (usually consulting firms and data vendors) are invited because the insurance company itself does not have any historical data on this new market. So, this is when the non-trivial MTPT procedure described in our work may apply given that the insurance company is only able to collect information about non-sensitive attributes for potential consumers in the new market. So, the insurance company may have to send this information ($T(X)$) to another party (a vendor) to do pricing since terms and conditions may restrict the vendor to only internally use the collected sensitive attributes.
> > >
> > > Case 2: "Second, the insurer collects information on both non-sensitive and sensitive attributes and sends this information to a third-party vendor to execute the pricing algorithm." This often happens in mid-size or smaller insurance companies where they cannot afford a research team and support with the computation power needed. Therefore, insurance companies of this type heavily rely on third-party vendors and data service platforms. Therefore, most of the time they first buy industry-wise data and process it with credibility techniques and then send the processed data ($T(X), S$) to the data service platform (i.e. Data Robot etc.) and do the pricing on the data service platform instead ($T, S$ both can be based on data transmission security purpose, but S is most likely already noisy when bought from the vendor as discussed in the interpretation of noise in our introduction section). Our MTPT procedure also applies when this type of insurance companies prepare for new market development.
> > >
> > > To summary:
> > >
> > > $\textbf{Q3-1:}$ "I am unsure of the need to send information if they can do pricing themselves."
> > >
> > > $\textbf{A3-1:}$ Ideally, the insurance company would want to do this on its own. However, the reality is that only a handful of tier-1 companies are capable of doing this for only most of the time. There are times where even they are not capable of doing this (mentioned in Case 1).
> > >
> > > $\textbf{Q3-2:}$ "Additionally, for the first case, I think this would contradict with the outlined procedure in 4.1. Would that mean the insurer doesn't do T(X), and h is performed on X?"
> > >
> > > $\textbf{A3-2:}$ Grouping is often done in rating variables such as age, number of children in the household, etc. So, one can consider such data processing as obtaining a $T(X)$ from the raw data. In a more general sense, T(X) is essentially considered as some sort of "feature engineering." when there is no data transmission to outside parties. So, it is up to the insurer whether they would do some data transformation before fitting the model and of course, directly performing $h$ on $X$ is a valid choice as $T(X) = X$ in this case.
> > >
> > > $\textbf{Q3-3:}$ "Lastly, since the insurer performing $h$ seems cheaper, less noisy, probably fairer, etc., than sending to TTP, even in the absence of sensitive attributes, I am still unsure of how the multi-party setting would be an incentive to the insurer."
> > >
> > > $\textbf{A3-3:}$ Yes, it has lots of advantages for the insurer to do pricing on its own. However, as mentioned in Case 2, most of the insurers on the market are not capable of doing so and in fact, nearly all the major pricing work is done at a data service platform for these mid-size or smaller insurers.

---

> > > > ### Comment · Reviewer_rqKi · 2023-11-21
> > > >
> > > > I appreciate the authors for addressing the raised questions and followup concerns, and revising the paper. I have decided to increase my score.

---

> > > > > ### Author Response · Authors · 2023-11-21
> > > > > **Thanks You Note**
> > > > >
> > > > > We again thank reviewer rqKi for all the effort and time spent reviewing our revisions. Your feedback, concerns, and suggestions have greatly helped us to improve the quality of our work, and at the same time, we appreciate the improvement of the score.

---

### Official Review · Reviewer_BixT · 2023-11-01

**Soundness:** 3 good
**Presentation:** 2 fair
**Contribution:** 3 good
**Rating:** 6
**Confidence:** 2

**Summary:**

This paper studies the problem of fair pricing in insurance. The insurance industry pursues actuarial fairness, a concept distinctive from the more commonly studied algorithmic fairness, and there lacks effective methods for designing pricing models that are actuarial fair. Motivated by this research gap, this paper proposed a method to train actuarial fair models for the practical scenario where an insurer has access to non-sensitive attributes, and a trusted third-party (TTP) partner has access to the corresponding privatized sensitive attributes. The proposed training method only requires access to privatized sensitive attributes via the TTP. The authors demonstrated the validity of their method by deriving relevant statistical guarantees and showing empirical effectiveness on an income prediction task.

**Strengths:**

This paper studied an important practical challenge of training fair ML model when the protected attributes are not readily available. The research problem has practical potentials as it is directly motivated by real-world insurance pricing. Along with the proposed algorithms, the authors provided solid theoretical results about the statistical guarantees for their performance.

**Weaknesses:**

This paper focused on ‘actuarial fairness’ definition formulated in an earlier paper. It is unclear whether this formulation is practical, and whether it is a broadly accepted formulation. Further literature review on its usage and potentials in practice will be helpful. On a related note, the term ‘actuarial fairness’ was used in the introduction paragraph without defining what it is. While I understand it was formulated later in the mathematical definition, it will be helpful to see how the insurance industry defines ‘actuarial fairness’ on the conceptual level first.

I also found it difficult to pinpoint what is novel in the paper. One motivation mentioned in the beginning of the paper is that the difference between actuarial fairness and other conventional algorithmic fairness notions calls for new fair algorithms, but it is unclear why a fair algorithm designed under privacy considerations for a conventional fairness concept would not work. It seems that the difficult comes from the unavailable sensitive attribute, but this is not an issue unique to the insurance pricing application. In addition, for the derivation of theoretical results, it would be useful to know whether and how the fairness or the noise or the multi-party training flow leads to challenges.

**Questions:**

1.	In insurance pricing, are there any popular fairness definitions that are already used in pricing mechanisms?

2.	How restrictive are Assumptions A and B in Section 4.3?

3.	What is the interpretation of noise in this context?

4.	The algorithms consider that only the protected attributes are sensitive, hence are stored with the third party. Is it reasonable to also consider non-protected attributes (in terms of fairness) are also not readily accessible to the insurer, but need to be obtained via a third party? If so, can the algorithms be generalized?

---

> ### Author Response · Authors · 2023-11-17
> **Respond to Reviewer BixT**
>
> We thank the reviewer for acknowledging the motivation of the work, namely its practical potential in solving real-world insurance pricing problems. We also thank the reviewer for carefully reading our paper and detailed comments. We appreciate all the feedback and questions the reviewer raised. In the following, we carefully respond to each of the concerns and suggestions that the reviewer raised.
>
> $\textbf{Q1:}$ "This paper focused on ‘actuarial fairness’ definition formulated in an earlier paper. It is unclear ...... mathematical definition, it will be helpful to see how the insurance industry defines ‘actuarial fairness’ on the conceptual level first."
>
> $\textbf{A1:}$ We first present the definition of actuarial fairness: The premium is considered actuarially fair if it accurately reflects the expected cost of the coverage provided to the policyholder.  It is a practical and broadly accepted formulation by the insurance industry. However, in recent years regulators have begun to question whether an actuarially fair premium should discriminate against policyholders based on sensitive attributes. As a result, insurers are required to demonstrate that premiums are not discriminative w.r.t. sensitive attributes. Under this backdrop, we focused on the discrimination-free premium, a concept recently proposed in the actuarial science literature. The discrimination-free premium satisfies the notion of fairness from a causal inference perspective [1]. It is free from both direct and indirect discriminations linked to sensitive attributes.
>
> $\textbf{Q2:}$ "I also found it difficult to pinpoint what is novel in the paper. One motivation ......  whether and how the fairness or the noise or the multi-party training flow leads to challenges."
>
> $\textbf{A2:}$ In our work, we employed the concept of discrimination-free premium for pricing insurance contracts. As the reviewer noted, the notion of fairness in this context is different from the concept of fairness in the conventional algorithm fairness literature. However, the discrimination-free premium is a conceptual framework and is not immediately applicable in practice due to regulatory constraints. More precisely, due to regulatory requirements, insurance companies are either prohibited from directly accessing sensitive attributes or are limited to accessing only a noised version of the sensitive attributes. Our research aims to introduce a methodology enabling insurers to derive discrimination-free premiums by effectively addressing the challenges imposed by regulatory requirements. In terms of how noise leads to challenges in the derivation of theoretical results, we added an empirical study on the robustness of Risk-LDP against the noise rate estimation error. A high-level result of the study is that Risk-LDP is robust against estimation error even if the error is large when $\pi$ is far away from $\frac{1}{|\mathcal{D}|}$. However, it suffers from both underestimation (more severe) and overestimation even if the error is small when $\pi$ gets close to $\frac{1}{|\mathcal{D}|}$. Please see Appendix E for figures and a more detailed discussion.
>
>
>
>
> [1] Mathias Lindholm, Ronald Richman, Andreas Tsanakas, and Mario V. Wuthrich. Discrimination-free
> insurance pricing. SSRN Electronic Journal, 2020. doi: 10.2139/ssrn.3520676.

---

> > ### Author Response · Authors · 2023-11-17
> > **Respond to Reviewer BixT**
> >
> > $\textbf{Q3:}$ "In insurance pricing, are there any popular fairness definitions that are already used in pricing mechanisms?"
> >
> > $\textbf{A3:}$ As explained in some parts of $\textbf{A1, A2}$, insurers are obligated to demonstrate actuarial fairness in their premiums. However, it is only in recent years that regulators have become increasingly concerned about sensitive attributes, particularly indirect discrimination based on these attributes. Therefore, discrimination-free pricing is a relatively new topic in the actuarial science literature. The discrimination-free premium definition our work follows (Definition 3.3) satisfies the notion of fairness from a causal inference perspective [1]. It is free from both direct and indirect discriminations linked to sensitive attributes as mentioned in $\textbf{A1}$.
> >
> > [1] Mathias Lindholm, Ronald Richman, Andreas Tsanakas, and Mario V. Wuthrich. Discrimination-free insurance pricing. SSRN Electronic Journal, 2020. doi: 10.2139/ssrn.3520676.

---

> > > ### Author Response · Authors · 2023-11-17
> > > **Respond to Reviewer BixT**
> > >
> > > $\textbf{Q4:}$ "How restrictive are Assumptions A and B in Section 4.3?"
> > >
> > > $\textbf{A4:}$
> > >
> > > $\textbf{Restrictions on Assumption A:}$ The restriction of Assumption A relies on the type of generator (which will influence the tail distribution of $\hat{\pi}$) and the number of data within each group (which will influence the accuracy of $\hat{\pi}$). The condition in Assumption A is equivalent to:
> > > $$
> > > \mathbb{P} \left( \frac{\left( 1 - \frac{1}{|\mathcal{D}|} \right)^2}{t} > \left|\hat{\pi} - \frac{1}{|\mathcal{D}|}\right| \right) \le \exp(\frac{-t}{K}),
> > > $$
> > > when $K > 0$ is a constant.
> > >
> > > Generally speaking, this assumption holds if $\hat{\pi}$ is inverse exponential distributed
> > > with a translation of $\frac{1}{|\mathcal{D}|}$, or having a lighter tail than the inverse exponential distribution that is
> > > $$
> > > f_{\hat{\pi}}(t) \le \frac{1}{K(t-|\\mathcal{D}|)^2}\exp(-\frac{1}{K|t - \frac{1}{\mathcal{D}}|}),
> > > $$
> > > when $t$ is close to $\frac{1}{|\mathcal{D}|}$, where $f_{\hat{\pi}}(t)$ is the pdf of $\hat{\pi}$. Especially, since a bounded distribution is also sub-exponential, if $|\hat{\pi} - \frac{1}{|\mathcal{D}|}| > \epsilon$, for some $\epsilon > 0$ condition is also satisfied. This will happen when the number of data within groups ($m$) is sufficiently large and $\pi - \frac{1}{|\mathcal{D}|}$ is large enough.
> > >
> > > $\textbf{Restrictions on Assumption B:}$ For Assumption B, the condition is equivalent to $\mathbb{E}[\frac{1}{\hat{\pi} - 1/|\mathcal{D}|}]$. Therefore, the closer $\pi$ and $\frac{1}{|\mathcal{D}|}$ is the more accuracy of $\hat{\pi}$ is needed to suffice this assumption.

---

> ### Author Response · Authors · 2023-11-17
> **Respond to Reviewer BixT**
>
> $\textbf{Q5:}$ "What is the interpretation of noise in this context?"
>
> $\textbf{A5:}$ In our method, the noise in sensitive attributes can arise in various scenarios including but not limited to:
> 1) Data collection mechanisms: In the data collection, whether conducted by the insurer or a third party, privacy mechanisms are employed as filters to encourage consumers to provide relevant information. These mechanisms introduce a degree of distortion to protect individual privacy.
>
> 2) Measurement errors: Sensitive attributes contain errors stemming from inaccuracies in the information provided by policyholders. This includes instances where policyholders furnish inaccurate information in sensitive attributes, intentionally or unintentionally.
>
> 3) Privatization for data transmission security: Sensitive attributes undergo privatization to ensure data transmission security. This may happen during transmissions from third parties to insurers or vice versa. The privatization process adds a layer of security but introduces noise in the sensitive attributes.
>
> Note that the above interpretations of noise are also included in the partially rearticulated introduction section.

---

> ### Author Response · Authors · 2023-11-17
> **Respond to Reviewer BixT**
>
> $\textbf{Q6:}$ "The algorithms consider that only the protected attributes are sensitive, hence are stored with the third party. Is it reasonable to also consider non-protected attributes (in terms of fairness) are also not readily accessible to the insurer, but need to be obtained via a third party? If so, can the algorithms be generalized?"
>
> $\textbf{A6:}$ Under the insurance pricing context, sensitive attributes refer to rating variables that regulators prohibit insurers from using to discriminate policyholders in pricing. For generalization to non-sensitive attributes, the answer is $\textbf{Yes}$. The proposed algorithm is readily applicable when non-sensitive attributes originate from third-party sources. $\textbf{However}$, its practical value becomes less evident, as it is common practice for insurers to acquire additional policyholder information through third parties such as credit reports.

---

> ### Comment · Reviewer_BixT · 2023-11-20
> **Thank you for the responses**
>
> I would like to thank the authors for the detailed response. The re-written introduction section is much clearer at explaining the fairness requirement in insurance, what makes attaining it difficult and this paper's new contribution.

---

> > ### Author Response · Authors · 2023-11-20
> > **Update in Assumption B**
> >
> > I would like to first thank the reviewer for the time and effort spent on reviewing our revision. We hope we have addressed the concerns and questions the reviewer raised in the earlier discussion. Inspired by Reviewer rqKi's question on the alternative when assumption B does not hold, we relaxed the exact unbiasedness assumption where some perturbation on the unbiasedness is allowed. Based on the new assumption B, we slightly modified Theorem 4.5 and the same result still holds in general up to some constant.
> >
> > Further, If our responses have addressed your concerns and questions, we would appreciate it a lot if you could please improve the score. Thank you once again for your valuable contribution to improving our work and we look forward to hearing your opinions.

---

> ### Author Response · Authors · 2023-11-23
> **Looking forward to Your Feedback**
>
> Dear Reviewer BixT,
>
> We apologize for reaching out again during this busy period. But with less than 12 hours remaining for the rebuttal, your feedback is crucial for our work. To better address the questions on the restrictiveness of assumption B in Theorem 4.5. We'd like to again point out an update on assumption B of Theorem 4.5, where the assumption on exact unbiasedness of $\hat{C}_{1,k}$ is relaxed and the same result in Theorem 4.5 still holds (please see Section 4.3, Appendix C.3, and Appendix G.4 for a more detailed discussion). We believe this relaxation on assumption B of Theorem 4.5 makes our statistical guarantees on our algorithm much more informative in practice.
>
> Further, if our responses have addressed your concerns and questions, we would appreciate it a lot if you could please improve the score. Thank you once again for your valuable contribution to improving our work and we look forward to hearing your opinions soon.

---

### Official Review · Reviewer_m8tn · 2023-11-05

**Soundness:** 1 poor
**Presentation:** 2 fair
**Contribution:** 1 poor
**Rating:** 5
**Confidence:** 4

**Summary:**

This paper discusses a practical method to produce 'discrimination-free prices' for a regression task with a finite number of sensitive attributes. The method essentially consists on training a separate regression model for each sensitive task, then aggregate the predictions according to some predefined (sensitive) group marginal. To introduce some measure of privacy into the model, the sensitive attributes of each sample are shared using randomized response.

**Strengths:**

The method itself is exceedingly simple to implement.

**Weaknesses:**

One major concern for me is the novelty of the algorithm, since it amounts to learning a per-sensitive-group regression model and  (weighted) averaging.

The other large concern relies on the claims that the proposed algorithm is differentially private. I think the authors should specify this claim more precisely, maybe by stating that its assumed x, y are public knowledge and that the model is therefore private wrt only the sensitive attribute.

**Questions:**

What are the formal privacy guarantees for the trained model, given that the private release mechanism is applied only on the sensitive attributes and not on the entire sample.

---

> ### Author Response · Authors · 2023-11-16
> **Response to Reviewer m8tn**
>
> We thank the reviewer for acknowledging the easiness of implementation of our proposed algorithm. We value the constructive feedback that the reviewer provided.
>
> In the following, we carefully respond to each of the concerns that the reviewer raised.
>
> $\textbf{Q1:}$ "One major concern for me is the novelty of the algorithm, since it amounts to learning a per-sensitive-group regression model and (weighted) averaging.", "What are the formal privacy guarantees for the trained model, given that the private release mechanism is applied only on the sensitive attributes and not on the entire sample."
>
> $\textbf{A1:}$ It appears that there was some misperception of our work. The two-step procedure that the reviewer outlined serves as a conceptual framework designed to derive a discrimination-free premium that satisfies the notion of fairness in the insurance sector. However, this conceptual framework is not immediately applicable in practice due to the regulatory constraints. More precisely, due to regulatory requirements, insurance companies are either prohibited from directly accessing sensitive attributes or are limited to accessing only a noised version of the sensitive attributes. Our research aims to introduce a methodology enabling insurers to derive discrimination-free premiums by effectively addressing the challenges imposed by regulatory requirements. $\newline$
>
> $\textbf{Q2:}$ "The other large concern relies on the claims that the proposed algorithm is differentially private. I think the authors should specify this claim more precisely, maybe by stating that its assumed x, y are public knowledge and that the model is therefore private wrt only the sensitive attribute."
>
> $\textbf{A2:}$ As mentioned in $\textbf{A1}$, the focus of our work is to introduce a methodology that enables insurers to derive discrimination-free premiums under challenges imposed by regulatory bodies. In Section 4.2, a brief introduction to LDP is given, and emphasis is also given that the privacy mechanism (LDP) is enforced only on the sensitive attribute $D$ during data collection of the trusted third party. All of our results are consistent with the fact that noise is introduced only to $D$ throughout the rest of our work. Further, LDP is only considered under the scenario where the data collector (trusted third party in our context) employs such privacy mechanism to encourage consumers to provide information about sensitive attributes (please refer to the noise interpretation in Section 1).

---

> > ### Comment · Reviewer_m8tn · 2023-11-22
> > **Thank you for the responses**
> >
> > Thank you for the response. The new draft is clearer on why this approach is warranted. I have revised my grade up

---

> > > ### Author Response · Authors · 2023-11-22
> > > **Thanks You Note**
> > >
> > > We again thank reviewer m8tn for all the effort and time spent reviewing our revisions. Your feedback, concerns, and suggestions have greatly helped us to improve the quality of our work, and at the same time, we appreciate the improvement of the score.

---

> ### Author Response · Authors · 2023-11-20
> **Looking Forward to your Feedback**
>
> Dear Reviewer m8tn,
>
> We would like to first thank you for the time and effort you spent reviewing our work. Your insights and feedback are very valuable for us to improve the quality of our work.
>
> We hope our responses addressed your concerns and questions. If you have any further questions or concerns, we are more than willing to provide any further information or clarification.
>
> If our responses have addressed your concerns, we would appreciate it a lot if you could improve the score. Thank you once again for your valuable insights and feedback which contributed a lot to improving our work. Looking forward to your feedback.

---

> ### Author Response · Authors · 2023-11-21
> **Looking Forward to Your Feedback**
>
> Dear Reviewer m8tn:
>
> We apologize for reaching out again during this busy period. But with only two days remaining for the rebuttal, your feedback is crucial for our work. If you have any further concerns, we are more than willing to provide further information or clarification. If our responses have addressed your concerns, we would appreciate it a lot if you could please improve the score. Thank you once again for your valuable contribution to improving our work and we look forward to hearing your feedback soon.

---

> ### Author Response · Authors · 2023-11-22
> **Looking Forward to Your Feedback**
>
> Dear Reviewer m8tn:
>
> We apologize for reaching out again during this busy period. But with less than 24 hours remaining for the rebuttal, your feedback is crucial for our work. If you have any further concerns, we are more than willing to provide further information or clarification. If our responses have addressed your concerns, we would appreciate it a lot if you could please improve the score. Thank you once again for your valuable contribution to improving our work and we look forward to hearing your feedback soon.

---

### Author Response · Authors · 2023-11-16
**Summary of Updates**

We express our gratitude to all reviewers for their insightful feedback and valuable suggestions, which have significantly contributed to the improvement of our paper.

In summary, our research aims to introduce a methodology enabling insurers to derive discrimination-free premiums by effectively addressing the challenges imposed by regulatory requirements.

Most of the reviewers find our study interesting (all reviewers), well-motivated (Reviewer BixT; Reviewer rqKi), and well-presented (Reviewer rqKi).

In response to their comments, we have carefully revised our paper and provided a detailed summary of these revisions in Appendix A. All the revisions of our paper are marked in blue color.

In the revision, we included more background on the current challenges faced by fair insurance pricing to make it easier for the audience to track the novelty and value of our work.

We address individual questions from each reviewer as follows:
1) We address Reviewer m8tn's question on privacy guarantees in our partially rearticulated introduction section.
2) We address Reviewer BixT's question on actuarial fairness, noise interpretation, and generalization on non-sensitive attributes in our partially rearticulated introduction section.
3) We address Reviewer BixT's question on restrictions on assumptions A and B in Appendix C.

Additionally, we also address questions and concerns that are not listed in the Question section, directions for these can be found in a detailed summary of updates in Appendix A.

---

### Meta-Review · Area_Chair_QRSr · 2023-12-10

**Metareview:**

The paper aims to mitigate discrimination in the regression task of (insurance) pricing when sensitive attributes are privatized. The authors have done a convincing job of motivating their paper and addressing many of the issues raised by the reviewers. However, I believe that most of the reviewers' criticisms (especially regarding contributions and novelty) are a consequence of a limited overlap between the paper and the ICLR call for papers. I strongly believe that the paper presents an interesting line of research and results that would be much more appreciated in a more appropriate venue (e.g. FAccT).  The paper studies an important issue (unfairness) in a specific application of machine learning, while the scope of ICLR is the general development of machine learning (and, of course, its societal considerations).

**Justification For Why Not Higher Score:**

I believe the problem of this paper is that it is not a good fit in scope for ICLR.

**Justification For Why Not Lower Score:**

N/A

---

### Decision · Program_Chairs · 2024-01-16

Reject